# The BrightEyes-TTM as an open-source time-tagging module for democratising single-photon microscopy

Alessandro Rossetta[1,2,3,8], Eli Slenders[1,8], Mattia Donato ®[1,8], Sabrina Zappone ®[1,3], Francesco Fersini ®[1,3], Martina Bruno[4,5], Francesco Diotalevi[6], Luca Lanzanò[2,7], Sami Koho[1], Giorgio Tortarolo[1], Andrea Barberis[4], Marco Crepaldi ®[6], Eleonora Perego ®[1] & Giuseppe Vicidomini ®[1] ✉

Fluorescence laser-scanning microscopy (LSM) is experiencing a revolution thanks to new single-photon (SP) array detectors, which give access to an entirely new set of single-photon information. Together with the blooming of new SP LSM techniques and the development of tailored SP array detectors, there is a growing need for (i) DAQ systems capable of handling the high-throughput and high-resolution photon information generated by these detectors, and (ii) incorporating these DAQ protocols in existing fluorescence LSMs. We developed an open-source, low-cost, multi-channel time-tagging module (TTM) based on a field-programmable gate array that can tag in parallel multiple single-photon events, with 30 ps precision, and multiple synchronisation events, with 4 ns precision. We use the TTM to demonstrate live-cell super-resolved fluorescence lifetime image scanning microscopy and fluorescence lifetime fluctuation spectroscopy. We expect that our BrightEyes-TTM will support the microscopy community in spreading SP-LSM in many life science laboratories.

A revolution is happening in fluorescence laser-scanning microscopy (LSM): the radically new way in which the fluorescence signal is recorded with single-photon array detectors drastically expands the information content of any LSM experiment. For a conventional fluorescence laser-scanning microscope, an objective lens focuses the laser beam at a specific position in the sample, called the excitation or probing region. The same objective lens collects the emitted fluorescence and, together with a tube lens, projects the light onto the sensitive area of a single-element detector, such as a photomultiplier tube

(PMT). Depending on the type of experiment, the probing region is scanned across the sample or kept steady. For imaging, the sample is raster scanned. For single-point fluorescence correlation spectroscopy (FCS), the laser beam is kept steady, whereas for scanning FCS, the laser beam scans the sample repeatedly in a circular or linear manner. During the whole measurement, the detector generates a one-dimensional signal (intensity versus time) that the data acquisition (DAQ) system integrates within the pixel dwell time (for imaging) or the temporal bins (for FCS). Such a signal-recording process induces

[1]Molecular Microscopy and Spectroscopy, Istituto Italiano di Tecnologia, Via Enrico Melen 85, Genoa 16152, Italy. [2]Nanoscopy and NIC@IIT, Istituto Italiano di Tecnologia, Via Enrico Melen 85, Genoa 16152, Italy. [3]Department of Informatics, Bioengineering, Robotics, and Systems Engineering, University of Genoa, Via All'Opera Pia 13, Genoa 16145, Italy. [4]Synaptic Plasticity of Inhibitory Networks, Istituto Italiano di Tecnologia, Via Morego, 30, Genoa 16163, Italy. [5]Department of Neuroscience, Rehabilitation, Ophthalmology, Genetics and Maternal and Child Sciences, University of Genoa, Largo Paolo Daneo, 3, Genoa 16132, Italy. [6]Electronic Design Laboratory, Istituto Italiano di Tecnologia, Via Enrico Melen 85, Genoa 16152, Italy. [7]Department of Physics and Astronomy "Ettore Majorana", University of Catania, Via S. Sofia 64, Catania 95123, Italy. [8]These authors contributed equally: Alessandro Rossetta, Eli Slenders, Mattia Donato. ✉e-mail: giuseppe.vicidomini@iit.it

information loss: the signal from the photons is integrated by the detector regardless of the photons' positions on the sensitive area and their arrival time with respect to a particular event, such as the fluorophore excitation event. Notably, also other properties of light, such as the wavelength and the polarization, are typically completely or partially discarded.

Single-photon (SP) array detectors, when combined with adequate DAQ systems, can preserve most of this information. In particular, asynchronous read-out SP array detectors—consisting of a matrix of fully independent elements that can detect a single-photon with several tens of picoseconds timing precision—have made it possible to spatiotemporally tag each fluorescence photon, i.e., to record simultaneously at which position of the array (spatial tag) and at which delay with respect to a reference time (temporal tag) the photon hits the detector.

Currently, the spatial tags can be used in two ways: firstly, by placing a bidimensional detector array in the LSM image plane, the probing region can be imaged (Fig. 1a). Secondly, by dispersing the fluorescence across the long axis of a linear detector array, the spatial tags enable spectrally resolved recording of the probing region, i.e., the spatial tag encodes the wavelength of the photon. At the same time, by exciting the sample with a pulsed laser and recording the fluorescence photon arrival times with respect to the excitation events, the temporal tags (i.e., the time difference between the excitation event and the photon detection, also called the start–stop time) allow for sub-nanosecond time-resolved measurements, such as fluorescence lifetime or photon correlation. Furthermore, recording the photon arrival times with respect to the beginning of the experiment allows for microsecond intensity fluctuation analysis, e.g., for time-resolved fluorescence fluctuation spectroscopy (FFS).

In summary, these spatial, temporal, and spectral photon signatures have opened a series of advanced fluorescence imaging and spectroscopy techniques precluded (or made more complex) by

conventional single-element detectors. Recently, a new LSM architecture based on linear SP detectors has led to a revival of the combination of fluorescence lifetime and spectral imaging[1]—spectral fluorescence lifetime imaging microscopy (S-FLIM). At the same time, bidimensional SP array detectors have opened up new perspectives for image-scanning microscopy (ISM). ISM uses the information contained in the image of the probing region to reconstruct the specimen image with a twofold increase in spatial resolution and a higher signal-to-noise ratio (SNR) compared to conventional LSM[2–4]. Because bidimensional SP array detectors provide a sub-nanosecond time-resolved image of the probing region, ISM can be combined with fluorescence lifetime to create a super-resolution fluorescence lifetime imaging technique, called fluorescence lifetime ISM (FLISM)[5], or to trigger the implementation of a new class of nanoscopy techniques, namely quantum ISM[6,7]. The microsecond time-resolved images can also be used to implement (i) high information content FCS and, more generally, fluorescence fluctuation spectroscopy[8]—usually referred to as comprehensive-correlation analysis (CCA)[9]; and (ii) the combination of super-resolution optical fluctuation imaging with ISM[10]. Hereafter, we refer to these techniques with the collective term single-photon laser-scanning microscopy (SP-LSM).

Key elements for implementing SP-LSM are the SP array detector and the DAQ system. Although analog-to-digital converters (e.g., constant-fraction discriminators) allow the use of photomultiplier tube arrays as SP detectors, they introduce a significant amount of unwanted correlation into the measurements[11] and are outperformed by true SP detectors regarding the photon-time jitter/precision. An alternative to PMT-based SP array detectors is the AiryScan-inspired module in which the hexagonal-shaped multi-core fiber bundle is connected to a series of single-element single-photon avalanche diodes (SPADs)[6] instead of PMTs as in the conventional AiryScan module[12]. Clearly, this module is expensive and not scalable. True SP array detectors based on the well-established SPAD array

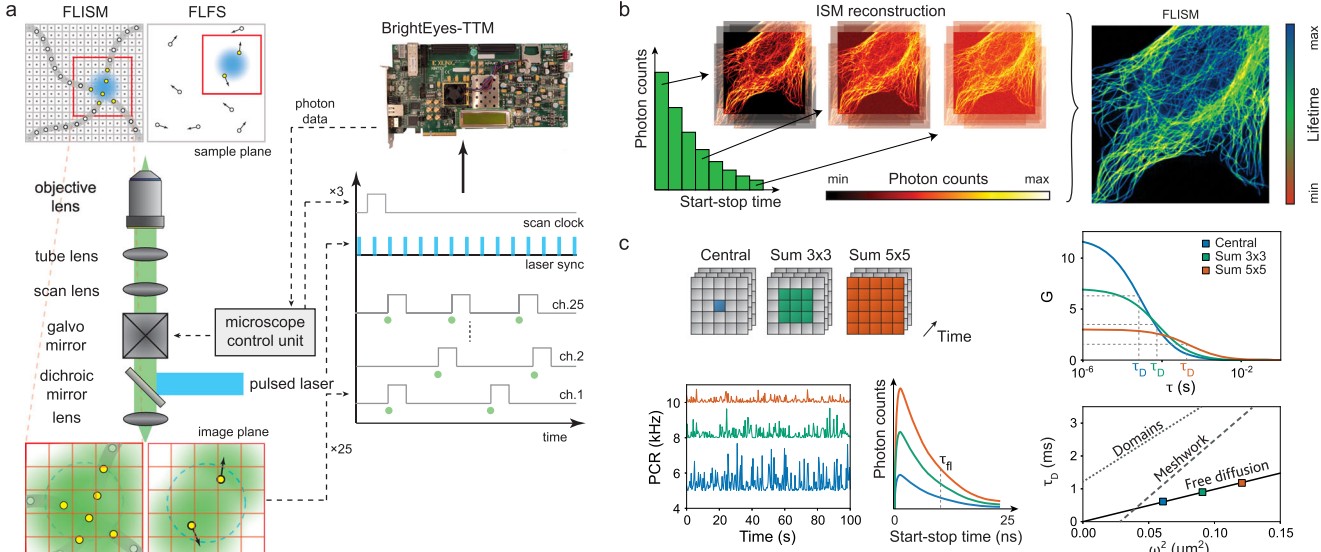

**Fig. 1 | Concepts of FLISM and FLFS. a** In FLISM, a pulsed laser beam is focused and scanned across the sample. For each position of the laser beam, the fluorescence is collected and focused onto a SPAD array detector. Every photon produces a pulse in one of the detection channels almost instantaneously. The BrightEyes-TTM measures the arrival time of the photon with respect to the last laser pulse and the pixel, line, and frame clock of the microscope. In single-point spot-variation FLFS, the laser beam is kept steady while the photon arrival times are measured. The movement of the fluorophores results in temporal fluctuations in the intensity trace. **b** A super-resolution fluorescence lifetime (FLISM) image can be reconstructed from the resulting 4D dataset ($x$, $y$, $t$, ch). For each time bin of the TCSPC

histogram, a super-resolved ISM image is reconstructed with the adaptive pixel reassignment algorithm. All the images are then recombined, and the fluorescence lifetimes are calculated for each pixel, resulting in the final FLISM image. **c** In spot-variation FLFS, the diffusion time as a function of the focal spot area is measured. The dimension of the focal spot can be changed by combining the photon traces coming from different detection channels. From the autocorrelations of the resulting intensity time traces, the diffusion times, and hence the diffusion modality (free diffusion, diffusion through a meshwork, or diffusion in a sample comprising isolated microdomains), can be found. Simultaneously, from the start–stop times, the fluorescence lifetime $\tau_{fl}$ is measured. PCR photon count rate.

technology[13,14] solved these limitations. In particular, asynchronous read-out SPAD array detectors have been tailored for SP-LSM applications[15–18]. These detectors have a small number of elements—but enough for sub-Nyquist sampling of the probing region— high photon-detection efficiency, low dark noise, high dynamics, high fill factor, low cross-talk, and low photon time-jitters.

Although the development of SPAD array detectors specifically designed for SP-LSM is gaining substantial momentum, no effort has been placed into developing an open-source data acquisition system able to (i) fully exploit the high-throughput and high-resolution photon-level information that these detectors provide; and (ii) offer flexibility and upgradability of the system. The lack of an open-source DAQ system may preclude a massive spreading of the above SP-LSM techniques and the emergence of new ones.

To address this need, we propose here an open-source multi-channel time-tagging module (TTM), called the BrightEyes-TTM, specifically designed to implement current and future fluorescence SP-LSM techniques. The BrightEyes-TTM has multiple photon- and reference channels to record at which element of the detector array, and when (with tunable precision) with respect to the reference events a single photon reaches the detector. The BrightEyes-TTM is based on a commercially available and low-cost field-programmable gate array (FPGA) evaluation board, equipped with a state-of-the-art FPGA and a series of I/O connectors that provide an easy interface of the board with the microscope, the SPAD array detector, and the computer. We chose an FPGA-based implementation to grant quick prototyping, easy updating, and adaptation: in particular, we envisage a module that can be updated—also remotely—by us or other groups to meet future requests from new SP-LSM techniques and SPAD array detectors.

We integrated the BrightEyes-TTM into an existing custom SP-LSM architectures equipped with a 5 × 5 SPAD array detector prototype or a commercial 7 × 7 SPAD array detector, and we performed FLISM imaging (Fig. 1b) on a series of calibration and biological samples, including living cells. Furthermore, for the first time, we demonstrated the combination of CCA with fluorescence lifetime analysis (Fig. 1c). This synergy opens to a new series of fluorescence lifetime fluctuation spectroscopy (FLFS) techniques able to provide a more complete picture of the biomolecular processes inside living cells. As proof-of-principle, we correlated the diffusion mode of eGFP protein with its fluorescence lifetime in live cells.

Despite the great potential of SP-LSM, we are aware that massive dissemination of this paradigm will be effective only if a broad range of laboratories will have access to the TTM, and potentially modify it according to their needs. For this reason, this manuscript provides detailed guidelines, hardware parts lists, and open-source code for the FPGA firmware and operational software.

## Results
### Multi-channel time-tagging module
The BrightEyes-TTM includes multiple (i.e., 25-channel for the current release and 49-channel for the next) fine (picosecond precision) time-to-digital converters (TDCs) to measure the start–stop times, and three coarse (nanosecond precision) TDCs to measure the relative delays between photon and clock signals. To characterize the performances of the fine TDCs, we used a test-bench architecture based on the SYLAP pulse generator. Independently, we validated the coarse TDCs directly by integrating the BrightEyes-TTM into different SP-LSM systems.

We measured the linearity of the fine TDCs—which expresses the deviation from the ideal behavior of the converter, by performing a statistical code-density test. We fed a fixed frequency signal (50 MHz) into the synchronization (SYNC) channel and a random signal into one of the photon channels (Channel #12). After accumulating several millions of photon events, we built the start–stop time histogram, also called the time-correlated single photon counting (TCSPC) histogram, which shows a differential nonlinearity (DNL) of $\sigma_{DNL} = 0.06$ least-

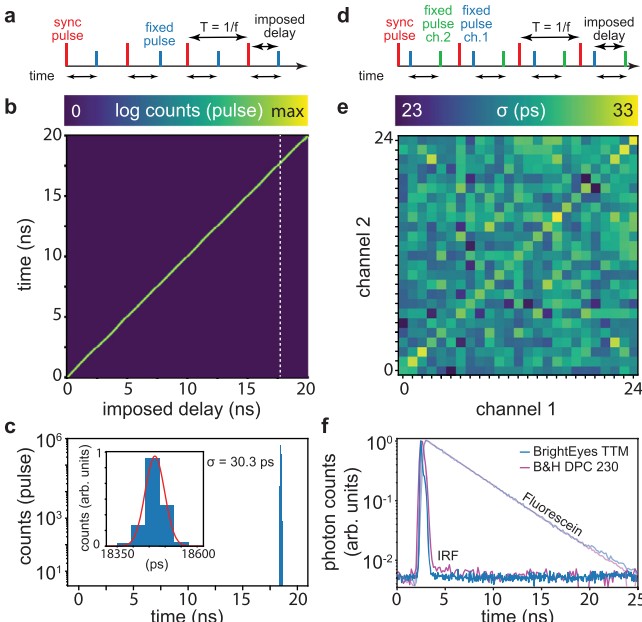

**Fig. 2 | BrightEyes-TTM characterization and validation. a–c** Single-shot precision experiment. **a** Temporal schematic representation: a fixed frequency SYNC clock signal and a synchronized but delayed (in a controlled way) signal. **b** Unified representation of the start–stop time histograms as a function of the imposed delay between the two signals. **c** Single start–stop time histogram for the delay denoted by the dotted white line in the middle panel. The inset shows a magnification of the histogram for a selected temporal interval, superimposed with the Gaussian fit (red line). **d, e** Dual-channel single-shot precision experiment. **d** Temporal schematic representation: a fixed frequency SYNC clock signal and a pair of synchronized signals (channel A and channel B). The delays between all three signals are fixed. **e** Jitter map for each pair of channels (here, 25 channels), i.e., error in the time-difference estimation between any two channels, measured as the standard deviation of a Gaussian fit of the error distribution. The diagonal of the map represents the sigma of the single-channel single-shot precision experiment. **f** Normalized impulse-response functions (dark colors) and fluorescein–water solution decay histograms (light colors) for the BrightEyes-TTM and DPC-230 multi-channel card. The instrument response functions (IRFs) represent the response of the whole architecture (microscope and DAQ) to a fast (sub-nanosecond) fluorescence emission. The full-width-at-half-max values are 250 ps for the DPC-230 card and 200 ps for the BrightEyes-TTM. The decay histograms are also typically referred to as start–stop time histograms or TCSPC histograms. All single-channel measurements were done with TTM channel #12, which received the photon signal from the central element of the SPAD array detector. All start–stop histograms have 48 ps granularity (bin width).

significant-bit (LSB) and an integral nonlinearity (INL) of $\sigma_{INL} = 0.08$ LSB—with LSB = 48 ps (Supplementary Fig. S1). Such low values of nonlinearity are negligible in a typical measurement of the photons' temporal distribution.

We then characterized the single-shot precision (SSP) of the fine TDC by repeatedly measuring a constant start–stop time interval. The SSP represents the standard deviation of the distribution around the mean value when a constant time interval is measured multiple times. We fed a fixed frequency (50 MHz) signal into the SYNC channel and a 30 times decimated synchronized (i.e., ~1.6 MHz) second signal—with a tunable fixed delay—into one of the photon channels (Channel #12). After accumulating several millions of sync-photon pairs, we built the start–stop time histogram, which in this case represents the distribution of the measurement error (Fig. 2a–c). By fitting the start–stop time histogram with a Gaussian function, we estimated a precision of $\sigma = 30$ ps (standard deviation of the fitted Gaussian distribution). We tuned the delay between the two signals across the whole temporal range of the fine TDC (here, 20 ns), and we observed a similar precision for all the imposed delays (Supplementary Fig. S2), confirming the

linearity of the fine TDC. We repeated the same SSP experiment for the other channels and obtained a similar precision (Fig. 2d, e). This SSP allows leveraging the photon-timing precision of the SPAD array detector.

Furthermore, we repeated the SSP experiment by feeding the same signal into a second photon channel. In this case, the delays between all three signals (Channel$_A$, Channel$_B$, and SYNC) are kept fixed, and we used the TTM to measure the delay between two photon channel signals. Similar to the start–stop time histogram, we built a histogram that reports the elapsed time between the two-photon channel signals, and we fit the distribution with a Gaussian function. We performed the experiment for all the possible channel pairs, and we obtained a $\sigma$ precision value ranging from 23 to 33 ps, depending on the channel pair (Fig. 2d, e).

Lastly, we checked the sustained photon rate of the BrightEyes-TTM by implementing the SPP experiment for increasing photon rates (from 100 kHz to 50 MHz), keeping the SYNC signal at 50 MHz. The module starts saturating at ~15 MHz for a single channel and at ~5 MHz when all 25 channels received simultaneously a photon signal, which corresponds to a total photon flux of 125 MHz (Supplementary Fig. S3).

After the test-bench measurements, we integrated the BrightEyes-TTM into a custom-built single-photon laser-scanning microscope equipped with a 5 × 5 SPAD array detector prototype and a picosecond pulsed diode laser. To measure the impulse-response function (IRF) of the system, we used a solution of fluorescein, saturated with potassium iodide to quench the fluorescence[19] (Fig. 2f). The relatively high full-width at the half-maximum value of the IRF (240 ps) is due to the convolution of the single-shot response (~30 ps) with the laser pulse width (>100 ps), the SPAD photon jitters (>90 ps), and the jitters/dispersion introduced by the optical system. We compared the IRF of the BrightEyes-TTM with the IRF of the DPC-230 commercial multi-channel time-tagging card measured on the same optical architecture. Notably, because of the poor time resolution (164 ps from the datasheet), the DPC-230 is not able to reveal the typical cross-talk effect of the SPAD array detector[17] which is visible in the BrightEyes-TTM as an additional bump (Fig. 2f). We used the two time-tagging systems to compare the decay distributions of a pure (not quenched) solution of fluorescein. The two TCSPC histograms show very similar shapes, Fig. 2f, which is confirmed by fitting them with a single exponential decay model: $\tau_{fl} = (3.97 \pm 0.04)$ ns and $\tau_{fl} = (3.99 \pm 0.01)$ ns, for the BrightEyes-TTM and DPC-230, respectively. To demonstrate the ability of the BrightEyes-TTM to work at different temporal ranges, we repeated the fluorescein experiment for different laser frequencies (80, 40, 20, 10, and 5 MHz). The TCSPC histograms do not show significant differences (Supplementary Fig. S4).

To test the BrightEyes-TTM for different fluorescence lifetime values, we measured the decay distributions of the quenched fluorescein solution for increasing concentrations of potassium iodide (Supplementary Fig. S5). Each measurement was analyzed by performing a single-exponential fitting of the TCSPC histogram and by phasor analysis. Phasor plots visualize the fluorescence lifetime by projecting the TCSPC histogram in a 2D coordinate system[20], which allows interpreting FLIM data without the need for fitting. The higher the potassium iodide concentration is, the higher the quenching will be, and thus the longer the measurement needs to be in order to accumulate good photon statistics. For this reason, the dark noise (which appears as an uncorrelated background in the TCSPC histogram) increases with the quencher concentration. The same effect appears on the phasor plot: because the decays follow a single-exponential function, caused by the collisional mechanism of the quenching, all points, regardless of the concentration, should lie on the universal semicircle. However, the uncorrelated background shifts the points toward the origin, since a lower signal-to-background ratio yields a higher demodulation.

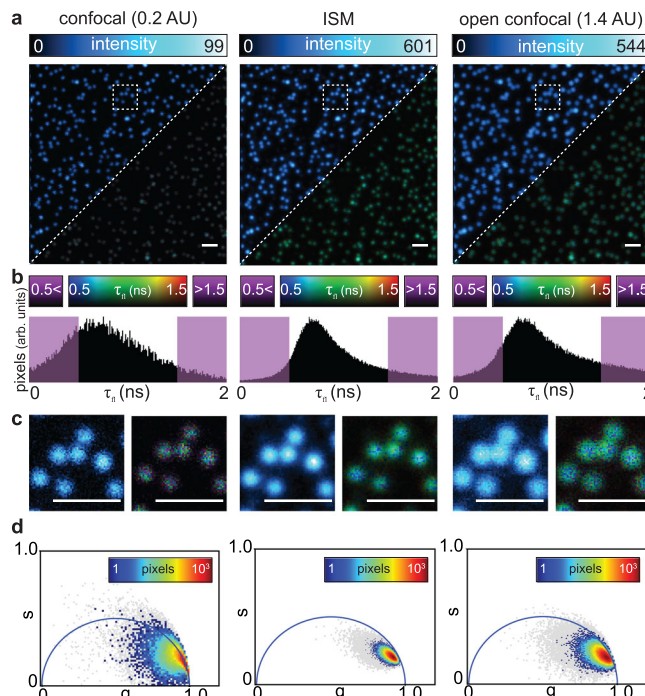

**Fig. 3 | BrightEyes-TTM for FLISM.** Imaging and FLISM analysis of 100 nm fluorescent beads with a custom-built single-photon laser-scanning microscope equipped with a 5 × 5 SPAD array detector prototype. **a** Side-by-side comparison of confocal (left, pinhole 0.2 AU), adaptive pixel-reassignment ISM (center), and open confocal (right, pinhole 1.4 AU). AU = Airy unit. Each imaging modality shows both the intensity-based image (top-left corner) and the lifetime image (bottom-right corner). A bidimensional look-up-table represents in the lifetime images both the intensity values (i.e., photon counts) and the excited-state lifetime values (i.e., $\tau_{fl}$). The intensity-based images integrate the relative 3D data ($x$, $y$, $\Delta t$) along the start–stop time dimension $\Delta t$. **b** Histogram distributions showing the number of pixels versus lifetime values—in violet lifetime values which fall out of the selected lifetime interval. The selected interval was chosen by visually inspecting the FLISM image. The same intervals were used for the confocal and open confocal data. The lifetime images report in violet the pixels whose lifetime is in this interval. **c** Zoomed regions in the white-dashed boxes, with the intensity panels renormalized to the maximum and minimum values. **d** Pixel intensity phasor plots, 5% and 10% thresholds respectively in gray and color. Pixel dwell time 100 μs. Scale bars 2 μm.

In conclusion, the BrightEyes-TTM offers a combination of single-shot precision, linearity, temporal range, number of channels, and sustained count rate which is suitable for measuring fluorescence lifetimes with state-of-the-art SPAD array detectors for LSM. It is worth noting that literature reports on different TDC implementations based on the same Kintex-7 FPGA family with superior characteristics (Supplementary Table S2). This aspect indicates the possibility of further improving the characteristics of our BrightEyes-TTM to match the expected enhancement in performances of the next SPAD array generations. In fact, it is important to observe that our current 25 channels TTM implementation (as well as the 49 channels implementation of the next BrightEyes-TTM release) uses a small portion of the FPGA resources available (Supplementary Fig. S6), thus offering room for implementing strategies to improve the overall characteristics of our TTM[21].

**Fluorescence lifetime image-scanning microscopy**
To demonstrate the ability of the BrightEyes-TTM in the context of SP-LSM imaging, we implemented FLISM on the same custom-made LSM used for the previous measurements. As with all LSM imaging techniques, FLISM requires acquiring the fluorescence photons in synchronization with the scanning system (e.g., galvanometric mirrors, and

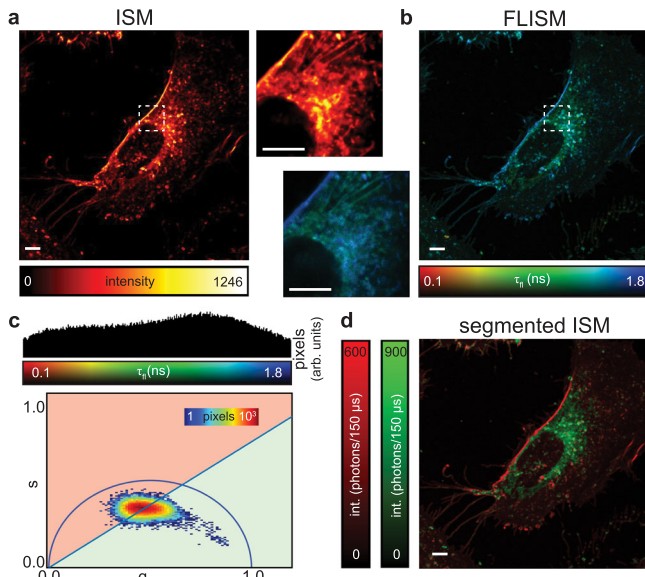

**Fig. 4 | Fluorescence lifetime image-scanning microscopy in live cells.**
**a** Intensity-based ISM image and **b** lifetime-based FLISM image of live HeLa cells stained with the polarity-sensitive fluorescent probe di-4-ANEPPDHQ. Side images depict the areas within the dashed white boxes. **c** Histogram distribution of the fluorescence lifetime values (top): number of pixels versus lifetime values. Pixel intensity thresholded phasor plots (bottom): number of pixels versus the polar coordinate (10% thresholds). **d** Phasor-based segmentation, i.e, images obtained by backprojection of points within the red (long lifetime, ordered membrane) and green (short lifetime, disordered membrane) area. Scale bars 5 μm. Pixel dwell time 150 μs. While here we show one cell, this experiment was independently performed two times on the same HeLa cell line. Setup: custom-built single-photon laser-scanning microscope equipped with a 5 × 5 SPAD array detector prototype.

piezo stages). We obtain this synchronization by measuring the start–stop times with respect to the clock signals typically provided by the scanning system (i.e., pixel/line/frame clocks). Since the scanning synchronization does not need high precision, we use the reference (REF) channels of the TTM, which have a coarse TDC with nanosecond precision (~4.2 ns). The current public BrightEyes-TTM has three REF channels (pixel, line, and frame), but additional channels can be implemented with minimal changes in the architecture. Having more REF channels would, for example, allow the recording of the photons in synchronization with a change in the excitation conditions, such as intensity, laser wavelength, or polarization.

Thanks to the synchronization signals, the stream of photons recorded by the BrightEyes-TTM leads to a 4D photon-counting image (ch, $x$, $y$, $\Delta t$), where $ch$ is the dimension that describes the element of the SPAD array, ($x$, $y$) are the spatial coordinates of the laser beam scanning system (in these experiments we recorded a single frame), and $\Delta t$ is the dimension of the TCSPC histogram (Supplementary Table S1). First, we used the adaptive pixel-reassignment (APR) algorithm[16,22] to reconstruct the 3D ($x$, $y$, $\Delta t$), high spatial resolution and high signal-to-noise ratio, image-scanning microscopy (ISM) intensity image. Next, we applied both conventional fluorescence lifetime fitting and phasor analysis to obtain the FLISM image ($x$, $y$) and the phasor plot ($g$, $s$), respectively. The side-by-side comparison of the intensity-based images of fluorescent beads clearly shows the optical resolution enhancement of ISM with respect to conventional (1.4 AU) imaging and the higher SNR with respect to confocal (0.2 AU) imaging (Fig. 3a, c and Supplementary Fig. S7). In the context of fluorescence lifetime imaging, the higher SNR leads to a higher lifetime precision, as depicted by the lifetime histograms and the phasor plots (Fig. 3b, d). We performed the same analysis on biological samples, fixed and live cells (Supplementary Fig. S8), which illustrates the flexibility of the

BrightEyes-TTM for different samples. Because the data collected by the BrightEyes-TTM need post-processing before being visualized, to obtain a real-time intensity-based image as a guide during FLISM experiments, we implemented in the BrightEyes-TTM a digital output signal which duplicates the signal of one photon channel. This duplicated signal (here the channel of the central element of the SPAD array) is sent back to the control system of the microscope, which performs real-time imaging (Supplementary Fig. S9).

We further validated the BrightEyes-TTM for a realistic application of fluorescence lifetime imaging, namely assessing the complex microenvironmental variation in a living cell (Fig. 4). In particular, we showed how the fluorescent polarity-sensitive membrane dye di-4-ANEPPDHQ allows visualizing ordered/disordered-phase membrane domains[16,23]. Monitoring the phase and phase changes of membranes is important since these domains affect many cellular membrane processes. While the fluorescent dye molecules in the plasma membrane show relatively long fluorescence lifetimes, denoting a high membrane order, dye molecules in the intracellular membrane show a shorter fluorescence lifetime, denoting membrane disorder. The difference between the two microenvironmental conditions can be highlighted with a phasor-based segmentation (Fig. 4c, d). We can follow the change in lifetime over time (S31, Supplementary Movie 1), which gives kinetic information on changes in the microenvironment of the fluorescent molecules. In general, this approach allows real-time tracking of changes in the structure and function of living systems, also allowing functional measurements such as Förster resonance energy transfer (FRET).

Finally, to confirm the versatility of the BrightEyes-TTM and to consolidate our vision about the future of SP-LSM, we demonstrated FLISM on a custom LSM setup equipped with a commercial 7 × 7 SPAD array detector (Supplementary Fig. S10). This detector uses the low-voltage-differential-signaling (LVDS) output standard instead of transistor-transistor-logic (TTL) as for the SPAD array prototype. Nonetheless, our BrightEyes-TTM can be used by upgrading the FPGA firmware and by substituting the I/Os daughter card with another one designed for the commercial 7 × 7 SPAD array detectors. In this work, we connected only the 5 × 5 central elements of the 7 × 7 SPAD array to the TTM.

## Fluorescence Lifetime Fluctuation Spectroscopy

To show the potential opened by the BrightEyes-TTM in the context of SP-LSM, we introduced fluorescence lifetime fluctuation spectroscopy (FLFS). Specifically, we developed two types of spot-variation FLFS, i.e., circular scanning FLFS and steady-beam (i.e., single-point) FLFS, and we used these techniques to probe the dynamics of fluorescent molecules both in vitro and in living cells. Because the BrightEyes-TTM provides both the photons' absolute times $t_{photon}$ (i.e., the delay of the photons with respect to the beginning of the experiment) and the photons' start–stop times, the diffusion dynamics and fluorescence lifetime of a molecule can be obtained simultaneously. Furthermore, because the detector array allows the simultaneous acquisition of fluorescence fluctuations in different detection volumes, spot-variation FCS can be straightforwardly implemented in a single experiment[8]: by measuring the diffusion time $\tau_D$ for different detection volumes, spot-variation FCS allows distinguishing between different molecular dynamics modes[24]. In addition, accessing the start–stop times allows filtering of the autocorrelation curves to mitigate artefacts such as detector afterpulsing.

As an example of circular scanning FLFS, we measured freely diffusing fluorescent nanobeads (Fig. 5). From the absolute times, we calculated the unfiltered autocorrelations for the central SPAD array detector element, the sum of the nine most central elements (called sum 3 × 3) and the sum over all elements except for the four corner elements (called sum 5 × 5). By scanning the probing region in a circle across the sample, the focal spot size $\omega_0$ and the diffusion time $\tau_D$ can

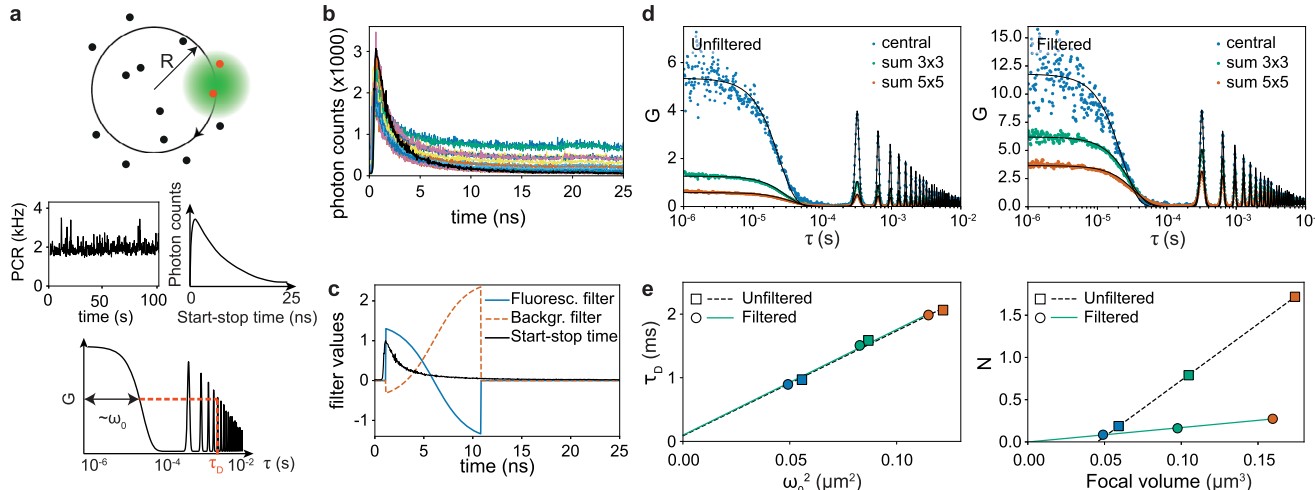

**Fig. 5 | BrightEyes-TTM for circular scanning FLFS on freely diffusing fluorescent beads. a** Schematic representation of the concept of circular scanning FLFS. A pulsed laser beam is scanned in circles of radius R (top panel) while both the absolute arrival times (center-left panel) and the start–stop times (center-right panel) are registered. The autocorrelation function of the intensity trace is calculated, from which the size of the focal spot $\omega_0$ and the diffusion time $\tau_D$ can be simultaneously extracted. **b** Start–stop time histograms for the different pixels, bin width 48 ps, total measurement time 226 s, central pixel in black. **c** Exemplary filter functions for the central pixel data. **d** Autocorrelations and fits for the central pixel, sum 3 × 3, and sum 5 × 5, for the unfiltered (left) and filtered (right) case. **e** Diffusion time as a function of $\omega_0^2$ (left) and average number of particles in the focal volume as a function of the focal volume (right). The corresponding diffusion coefficients are $(14.3 \pm 0.5)$ μm²/s (unfiltered) and $(14.0 \pm 0.4)$ μm²/s (filtered). The fitted particle concentrations are $(7 \pm 3)$/μm³ (unfiltered) and $(1.70 \pm 0.03)$/μm³ (filtered). Setup: custom-built single-photon laser-scanning microscope equipped with a 5 × 5 SPAD array detector prototype.

be derived from the same experiment[25], i.e., no further calibration measurement is needed to obtain $\omega_0$ (Fig. 5a). In all cases, we fitted the data with a model assuming a 3D Gaussian focal volume with the diffusion time, the particle concentration, and $\omega_0$ as fit parameters (Fig. 5d). The fitted diffusion coefficient, $(14.3 \pm 0.5)$ μm²/s, corresponds to the value that can be expected based on the diameter of the beads (estimated bead radius $r = 27$ nm). The diffusion law $\tau_D(\omega_0^2)$[26,27] confirms that the beads were freely diffusing: the dependency of $\tau_D$ is linear with a zero intercept (Fig. 5e, left panel). However, the fitted concentration of the beads sample does not scale proportionally to the focal volume (Fig. 5e, right panel, squares), indicating that the amplitudes of the autocorrelation functions are not correct. Using the start–stop time histograms (Fig. 5b), we calculated the filter functions which attenuate the detector afterpulsing and background signals (Fig. 5c). The TCSPC histogram of each channel was fitted with an exponential decay function with an offset, describing the fluorescence and the background, respectively. From the resulting fit, the filter functions, which assign a weight to every count, were calculated[28,29]. Calculating the autocorrelation functions of the weighted photon traces yields significantly altered amplitudes, whereas the diffusion times are mostly left unchanged. As a result, the filtered autocorrelations show both the expected behavior for the diffusion time and the concentration (Fig. 5e, circles). If the fluorescence signal is strong enough with respect to the background, filtering is not needed, as shown for freely diffusing goat anti-mouse antibodies conjugated with Alexa 488 (Supplementary Fig. S11).

Especially in a biological context, having simultaneous access to the lifetime and to diffusion information is useful not only for obtaining a robust spot-variation FCS analysis but also to correlate the mobility mode of the investigated molecules with their micro-environment or structural changes. In fact, by using probes sensitive to specific environment states or FRET constructs, the variation in the fluorescence lifetime can be linked to biomolecular changes[30]. As a proof-of-principle example, we used the BrightEyes-TTM to measure the diffusion of monomeric-eGFP inside living cells and correlated its mobility with its fluorescence lifetime (Fig. 6). Since our spot-variation FCS implementation allows extracting the mobility mode from a specific cell position[18], e.g., the measurement position for single-point

FCS, and within a specific temporal window (a few seconds), it allows revealing both spatial and temporal heterogeneity in the molecular diffusive behavior. With the BrightEyes-TTM, we can simultaneously perform ISM and FLISM on whole cells (Fig. 6a), and then select specific positions for performing FLFS. For each position, we use (i) the absolute arrival times to create the fluorescence time traces (Fig. 6c), and calculate the corresponding autocorrelations, (Fig. 6d) and (ii) the photon arrival times to create the start–stop histograms (Fig. 6b). Spot-variation FCS performed at different cell positions showed free diffusion dynamics but different diffusion coefficients, Fig. 6e, indicating mobility variability at the single cell level. However, the average diffusion coefficient, $D = (34 \pm 12)$ μm²/s, is well comparable with previous measurements of free eGFP in living cells[18,31] and with the expected diffusion value knowing its dimensions (MW = 27 kDa for monomeric-eGFP). We performed measurements in both the cell cytoplasm and the nuclei. Even if the eGFP molecules were not expected to be expressed in the cell nuclei, the (monomeric) eGFP molecules could diffuse into the nuclei due to their small dimension[32]. A slower diffusion of $(20 \pm 3)$ μm²/s was measured in the nuclei compared to $(35 \pm 10)$ μm²/s in the cytoplasm. This difference might be caused by the packed molecular environment of the nuclei. Repeating the analysis for consecutive time windows of 5 s each for a single position in the cell (here shown for the cytoplasm, circle shown in Fig. 6a), we find a temporal variability of the mobility mode (Fig. 6f). As a parameter for mobility, we calculated the ratio between the diffusion coefficients at different spatial scales, i.e., $D_{central}/D_{5\times5}$, which reflects the type of mobility, and we correlated it with the fluorescence lifetime, Fig. 6g. In this biological context, no differences in the fluorescence lifetime are expected over time, as eGFP is not specifically tagged to any protein. Moreover, the constant behavior of the fluorescence lifetime over time is proof of the viability of our system with biological samples for long acquisition times. On the other hand, the change in the diffusion modality suggests more complex protein dynamics, which might be caused by the heterogeneous cytosolic environment during physiological cellular processes. Similar to imaging, we took advantage of the BrightEyes-TTM output signal, which duplicates the central element photon channel, and we fed this signal into the control system of the microscope for a real-time display of the

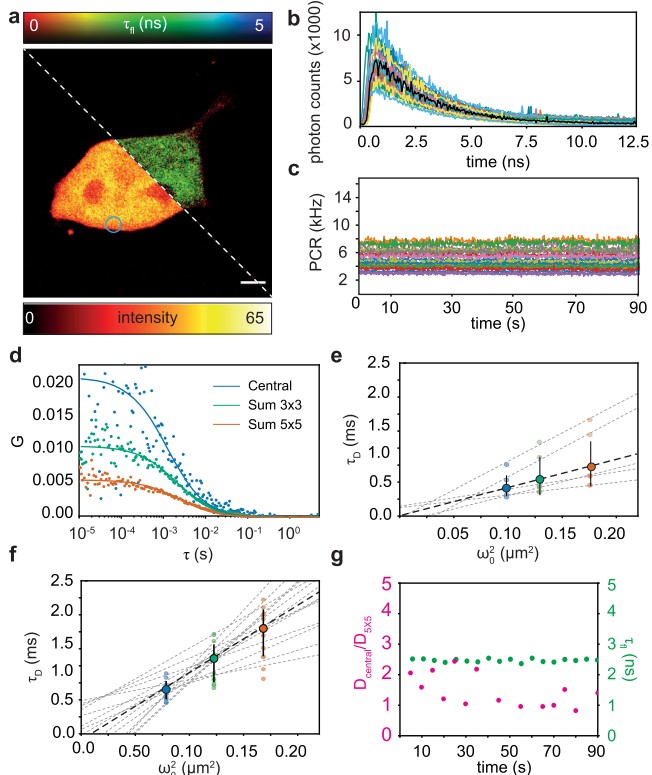

**Fig. 6 | Fluorescence fluctuation spectroscopy on living cells. a** ISM (bottom-left corner) and FLISM (top-right corner) images of a HEK293T cell expressing eGPF. **b**, **c** Start–stop time histograms (central pixel in black) and intensity time traces for all 25 channels of a 100 s FLFS measurement. The blue circle in **a** depicts the position in the cell where the measurement is performed. **d** Autocorrelation curves (scattered points) and fits (lines) for the central pixel, sum 3 × 3 and sum 5 × 5 curves obtained from (**c**). **e** Spot-variation analysis: the dashed black line represents the average ($D = 34 \pm 12$ μm²/s, $N = 5$) of the dashed light-gray lines. Each dashed light-gray line represents a different position within the same cell. **f** Spot-variation analysis as a function of the measurement time-coarse. Data from (**c**). The intensity time traces are divided into chunks of 5 s, each chunk is analyzed by means of spot-variation FCS and generates a dashed light-gray line. The dashed black line represents the average ($D = 32 \pm 5$ μm²/s, $N = 14$) of the dashed light-gray lines. Error bars in (**e**, **f**) represent standard deviations. **g** Ratio between the diffusion coefficients measured for the central pixel and sum 5 × 5 ($D_{central}/D_{5×5}$), overlapped with the fluorescence lifetime as a function of the measurement time-coarse. Data from (**f**). Scale bar 5 μm. Pixel dwell time 100 μs. The data were acquired from five different cells, in each cell multiple positions (from 3 to 5) were sampled. Setup: custom-built single-photon laser-scanning microscope equipped with a 5 × 5 SPAD array detector prototype.

intensity trace and the corresponding autocorrelation function. The autocorrelation obtained from the BrightEyes-TTM is identical to the curve obtained from the reference system (Supplementary Fig. S12).

## Discussion

We present a low-cost (<3000$) and multi-channel TTM-DAQ system specifically designed for fluorescence SP-LSM: a family of methods that leverage the possibility of SP array detectors to analyze the specimen's fluorescence photon-by-photon. We first validated the versatility of the BrightEyes-TTM by implementing super-resolution FLISM. Next, to demonstrate the perspectives opened by the BrightEyes-TTM, we introduced another SP-LSM technique, namely FLFS. We recently demonstrated that SPAD array detectors are the method of choice to implement CCA[8]. CCA combines in a single experiment a series of well-established fluorescence fluctuation spectroscopy approaches to

provide a global analysis for deciphering biomolecular dynamics in living cells. FLFS combines CCA with fluorescence lifetime analysis to further enhance the information content of a single experiment. We used FLFS to monitor the relationship between the diffusion mode of an eGFP-tagged protein and its fluorescence lifetime. In this context, we envisage the combination of FLFS with FRET sensors to open new opportunities for studying complex processes under physiological conditions[33].

We implemented the TTM architecture on an FPGA, mounted on a readily available development kit, thus providing design flexibility and upgradability and also allowing fast prototyping and testing of new potential BrightEyes-TTM release versions. Compared to general-purpose commercial TTMs, our current BrightEyes-TTM version provides similar characteristics (linearity, temporal range, and sustained count rate) or partially inferior characteristics (temporal precision) but is still ideally suited for SP-LSM with current SPAD array detectors (Supplementary Table S3). Indeed, because our TTM has been specifically designed for fluorescence analysis, temporal constraints can be relaxed at the benefit of (i) compatibility with existing LSM systems, (ii) the possibility to integrate other features, and (iii) scalability in terms of photon channels. The BrightEyes-TTM supports 25 channels (i.e., the optimal number of elements requested for a SP detector array to implement ISM) in a single module. All commercial TTMs require multiple synchronized modules to support 25 channels. Furthermore, we anticipate the implementation of a new open-source BrightEyes-TTM version capable of handling up to 49 photon channels, which are useful for other SP-LSM techniques, such as S-FLIM.

Whilst the time-tagging module implements the most informative DAQ system strategy, it requires transferring (to the PC) and storing a consistent amount of data. Thus, this approach is not practically scalable to the large (megapixel) SPAD cameras requested by wide-field microscopy[34]. Here, typically an application-specific integrated circuit (ASIC) TDC is implemented directly at the chip level for each SPAD element or cluster of SPAD elements[35], and the temporal information is pre-processed to reduce the amount of data to transfer. For example, during the measurement, the detector records the start–stop histogram, rather than the photon time-tags. However, also in the context of small SPAD array detectors, it could be important to reduce the data transferred to the PC. For this reason, the potential directions for the next developments in the BrightEyes-TTM project would be the migration to a new FPGA-development kit equipped with ARM-based processors. In this case, it might be interesting exploring next generation FPGAs with improved technological node (e.g., Ultrascale 20-nm), on which TDCs with superior performances are demonstrated[36].

The principal aim of this work is to democratize SP-LSM by giving any microscopy laboratory the possibility to update its existing LSM systems. Together with a TTM, the other important element in order to achieve our aim is the SPAD array detector. We demonstrate here the compatibility of the BrightEyes-TTM not only with a SPAD array detector prototype but also with a commercial SPAD array detector. Currently, two startups are selling SPAD array detectors for SP-LSM (Genoa Instruments and Pi-Imaging Technology), but the interest from the major microscopy companies for this technology is growing: The AiryScan from Zeiss has become very popular and its transition to a single photon-detector will be natural. On the other side, Genoa Instruments and Abberior Instruments already proposed integrating a SPAD array detector in their LSM product. Following this trend, we expect that also well-established detector companies will release a SPAD array detector specifically designed for SP-LSM.

The second aim of this work is to trigger the interest of the microscopy community and establish the BrightEyes-TTM as a common platform for further developments in the context of SP-LSM. We are fully aware that these aims can only be achieved if the microscopy

maker community[37] has full access to this device, thus we released the results of this work as open-source. The transition from conventional LSM to SP-LSM can pass through the implementation of standard FLIM with a single-element detector, such as a PMT or a SPAD. Thereby we also demonstrated the use of the BrightEyes-TTM to transform a commercial Nikon confocal LSM into a FLIM system (Supplementary Fig. S13), and to support well-established single-element SPADs for FLIM (Supplementary Fig. S14).

By implementing different low-precision reference signals, the TTM can collect photons in synchronization with many different optomechanical devices that could potentially contain, directly or indirectly, additional information about each photon. For example, polarization modulators and/or analyzers can help tag photons with an excitation and emission polarization signature. Acoustic-tunable filters and spectrometers can provide an excitation and emission wavelength signature.

We anticipate a future in which each single fluorescence photon will be tagged with a series of stamps describing not only its spatial and temporal properties, as shown in this work, but also its polarization and wavelength characteristics. A series of new algorithms—based on conventional statistical analysis or machine learning techniques, and data formats—will be developed to analyze and store such multi-parameter single-photon data-sets. As single-molecule techniques[38] have revolutionized cellular and molecular biology—by observing molecule-by-molecule and not as an ensemble, we expect that recording the fluorescence signal photon-by-photon, while preserving most of the encoded information, can provide similar outcomes.

Although we believe that the life-science community will receive the major benefits of the SP-LSM technology, we are convinced that SP-LSM will find many applications also in material sciences, in particular in the field of single-dot spectroscopy[39].

## Methods

### Time-tagging module architecture

In the context of SP-LSM, the goal of the BrightEyes-TTM is to tag every single photon that reaches the detector with a spatial signature and a series of temporal signatures (Supplementary Fig. S15). The temporal signatures are the delays of the photons with respect to specific events: (i) the fluorophore excitation events induced by a pulsed laser; (ii) the changes of some experimental condition (called here second-class or REF events), e.g., the start of the experiment or the position change in the probing region. Measuring the photon arrival time with respect to the excitation event (the so-called start–stop time) typically requires high temporal precision (higher than the SPAD array detectors photon-timing jitters, < 100 ps) and a temporal range larger than the pulse-to-pulse interval of the laser (10–1000 ns for 1–100 MHz laser pulse frequencies). For all second-class events, a nanosecond temporal precision is sufficient but the requested temporal range increases up to seconds. To meet these conditions, we implemented different time-to-digital converters (TDCs) for the different types of input signals. In particular, we implemented fine TDCs with picosecond precision for the photon and laser SYNC signal and coarse TDCs with nanosecond precision for the reference (REF) signals (Supplementary Fig. S16). Since most fluorescence applications do not require a subpicosecond temporal precision and we aimed to develop an open-source upgradable TTM, FPGA-based TDCs were the natural choice for the BrightEyes-TTM.

Within the FPGA-based TDC, the literature reports several different architectures[21]. Each architecture offers a compromise between different specifications (e.g., temporal precision, temporal range, temporal resolution, dead time, linearity, and FPGA resources). Using an FPGA, the delay between two events can be measured with a counter which simply counts the number of clocks between the two events. The counter approach yields a precision no better than the clock period, which is typically a few nanoseconds for low-cost devices

(as in our case) and—in principle—infinite temporal range. While this precision is suitable for measuring the start–stop time with respect to second-class events, the counter approach does not satisfy the request for the start–stop time assessment. For the delay with respect to the excitation event (fine TDC), a higher precision (10–100 picoseconds) is achieved by running the photon signal (i.e., the START signal) through a fast tapped-delay-line (TDL), and then measuring the position that is reached when the SYNC signal from the laser (i.e., the STOP signal) is received. The downside of the TDL approach is that the maximum delay measurable (i.e., the temporal range) depends on the length of the TDL, and thus on the available FPGA-resources. To keep the FPGA-resources low and to implement on the same architecture both a few coarse TDCs and one fine TDC for each SPAD array element, we implemented a sliding-scale interpolating TDC architecture[40] (Supplementary Fig. S17). This method uses a pair of tapped delay lines to measure with tens of picoseconds precision the start-stop time, combined with a free-running coarse counter to extend the temporal range. At the same time, the architecture uses the free-running coarse counter to measure the second class events with a few nanoseconds precision. In particular, the architecture combines $N+1$ ($N = 25$ photon channels in this implementation) tapped delay lines and a coarse counter at 240 MHz to obtain $N$ fine TDCs with tens of picoseconds precision (for the start-stop time of each photon channel), and $M$ coarse TDCs with a nanosecond precision ($M = 3$ reference channels in this implementation). For both TDCs, the temporal range is—in principle—limitless. Notably, each fine TDC uses a dedicated START tapped delay line, but shares with the other fine TDCs the STOP delay line; thus, the start-stop time resulting from each fine TDC is measured with respect to the same SYNC signal, which is what is typically needed in all SP-LSM applications. Furthermore, the high-frequency clock (240 MHz in our case) allows for keeping the delay lines short (slightly longer than the clock period, ~4.2 ns), thus reducing both the FPGA resources needed and the dead time of the fine TDC. Notably, the dead time of our TDC (~4.2 ns) is shorter than typical SPAD detector hold-off times (>10 ns) and typical pulsed laser excitation periods for fluorescence applications (12.5–50 ns).

By using two different tapped-delay lines for the START and STOP signals, the architecture ensures that the TDCs are asynchronous with respect to the clock counter. The asynchronous design reduces the nonlinearity problem of the FPGA-based TDC and auto-calibrates the tapped-delay lines (Supplementary Note 1 and Supplementary Fig. S17).

Here, we describe a single fine TDC. The parallelization to $N$ TDCs is straightforward, and the implementation of the coarse TDC comes naturally with the use of the coarse counter. Each fine TDC is composed of two flash TDCs, each one containing a tapped-delay line and a thermometer-to-binary converter (Supplementary Note 3 and Supplementary Fig. S18). The START delay line measures the difference $\Delta t_{START}$ between the START signal and the next active edge of the counter clock. Similarly, the STOP delay line measures the difference $\Delta t_{STOP}$ between the STOP signal and the next active clock rising edge. Thanks to the free-running coarse counter, the architecture is also able to measure (i) $n_{photon}$ and (ii) $n_{SYNC}$, corresponding to the number of clock cycles between the start of the experiment and (i) the photon signal or (ii) the SYNC signal. Given these values, the start–stop time $\Delta t$ is equal to $\Delta t_{STOP} - \Delta t_{START} + \Delta n \cdot \mathcal{T}_{sysclk}$, with $\Delta n = n_{SYNC} - n_{photon}$, and $\mathcal{T}_{sysclk} = 1/240$ MHz the clock period; the coarse absolute time $\hat{t}_{photon}$ is equal to $n_{photon} \cdot \mathcal{T}_{sysclk}$. Similarly, the nanosecond delay from the REF signal $\Delta \hat{t}_{REF}$ is equal to $(n_{(photon)} - n_{REF}) \cdot \mathcal{T}_{sysclk}$, with $n_{REF}$ the number of clock cycles between the beginning of the experiment and the REF signal. Since the TTM architecture can only provide integer values, a calibration is required to obtain the time values. In particular, the coarse counter provides the values $n_{photon}$ and $n_{SYNC}$, and the time-to-bin converters of the flash TDCs provide the values $\Delta T_{photon}$ and $\Delta T_{SYNC}$. After the raw data are transferred to the PC, the calibration is

used to calculate the time values $\Delta t_{photon}$ and $\Delta t_{SYNC}$ (Supplementary Note 1), and the application-dependent analysis software allows calculating the time interval between specific events (e.g., $\Delta t$, $\Delta \hat{t}_{REF}^{photon}$). This approach does not preclude applications in which some other notion of an absolute time is required, such as the absolute photon time (e.g., coarse $\hat{t}_{photon}$, or fine $t_{photon}$), rather than the time interval between two events.

The TTM architecture also contains hit filters (Supplementary Fig. S19), an event filter (Supplementary Fig. S20), and a data module in charge of preparing and transferring the data to the PC. The hit filter is a circuit used to shape and stabilize the input signal. The event filter is a data-input/output filter that reduces the data throughput rate by avoiding transmitting information when no photons are detected. The data-module buffers the data into a FIFO before being transferred to the PC. In its current implementation, data transfer is done with a USB 3.0 protocol, but the platform is compatible with other data communication protocols, such as a PCIe.

We implemented the BrightEyes-TTM architecture on a commercially available Kintex-7 FPGA evaluation board (KC705 Evaluation Board, Xilinx), featuring a state-of-the-art FPGA (Kintex-7 XC7K325T-2FFG900C) and, importantly for the versatility of the architecture, a series of hardware components (e.g., serial connectors, expansion connectors, and memories). In particular, the different serial connectors potentially allow for different data-transfer rates, and the expansion connectors allow compatibility with other detectors and microscope components (Supplementary Fig. S21).

To transmit the data to the PC via USB 3.0, the current TTM design uses an FX3-based board (Cypress SuperSpeed Explorer kit board, CYUSB3KIT) connected through an adapter card (CYUSB3ACC) to the LPC-FMC connector of the Kintex-7 evaluation board (Supplementary Fig. S21). To use the FX3, we developed a dedicated module in the FPGA. This module has a simple interface (essentially FIFO with a data-valid flag) for the data transmission, and it manages the FX3 control signals and the data bus. We designed the module to work with the FX3 programmed with the SF_streamIN firmware, which is part of the AN65974 example provided by Cypress. The total component cost for the BrightEyes-TTM is about $3000.

## Test-bench architecture and analysis

We performed the code-density test (Supplementary Note 2), the single-photon precision measurement and the saturation analysis by connecting the BrightEyes-TTM to a dedicated signal generator, named SYLAP (Supplementary Fig. S22). We implemented SYLAP on an FPGA evaluation board identical to the one used for the BrightEyes-TTM. The SYLAP architecture generates a fixed frequency clock, which we used to simulate the laser SYNC signal, and a synchronized pulse train, which we used to simulate the photon signal. The key features of the SYLAP are (i) the possibility of adjusting the delay from the clock and the pulse with a granularity of 39.0625 ps; (ii) the possibility of setting the clock period and the pulse duration with a granularity of 2.5 ns; (iii) the possibility of setting the number of clocks needed to generate a pulse, i.e., clock decimation. The native USB 2.0 serial port of the evaluation board allows configuring the SYLAP. The timing jitter between the clock and the pulse is 13 ps, measured with an oscilloscope (Keysight DSO9404A with 4 GHz bandwidth). Since both the BrightEyes-TTM and the SYLAP signal generator are based on the same type of board, we also implemented both systems on the very same FPGA. Importantly, in this case, we separated the clock domains and the clock sources of the BrightEyes-TTM and SYLAP projects. We performed a single-photon precision experiment to compare the stand-alone SYLAP configuration (i.e, two different boards connected with coaxial cables) with the internal configuration, and we did not observe substantial differences.

To perform the code-density test (Supplementary Note 2), we used the SYLAP clock signal as the SYNC signal and the TTL signal from

an avalanche photodiode (APD, SPCM-AQRH-13-FC, Perkin Elmer) as temporally uncorrelated photon signals. We illuminated the APD with natural light, maintaining a photon flux well below the saturation value of the detector. For the single-photon precision experiment, we used the clock and pulse signals from SYLAP. Specifically, to measure the precision for all photon channels, we took again advantage of the flexibility of the BrightEyes-TTM architecture: we implemented physical switches which allow simultaneously connecting a single input/ photon channel of the board to all photon channels. This feature gives the possibility to measure the same event with all photon channels. Finally, to analyze the sustained read-out rate of the BrightEyes-TTM, we performed the single-photon precision experiment for different clock decimations. We measured both the rate for a single active channel and for all channels active at the same time.

## Laser-scanning microscopes

**Optical architectures.** To test the BrightEyes-TTM, we used two custom-built laser-scanning microscopes and a commercial confocal laser-scanning microscope. We performed the majority of the FLISM and FLFS experiments with a custom system (Supplementary Fig. S23) previously implemented to demonstrate confocal-based FFS with a SPAD array detector prototype[8,18]. In particular, the microscope excites the sample with a triggerable 485 nm pulsed laser diode (LDH-D-C-485, PicoQuant) and records the fluorescence signal with a cooled BCD-based $5 \times 5$ SPAD array detector[17,18] or with a commercial single-element SPAD detector ($PD-050-CTC-FC, Micro Photon Devices). To test the BrightEyes-TTM with a commercial SPAD array detector, we used a second custom system (Supplementary Fig. S10a) previously implemented to introduce FLISM[16]. In particular, the microscope excites the sample with a triggerable 640 nm pulsed laser diode (LDH-D-C-640, PicoQuant) and records the emitted fluorescence with a $7 \times 7$ CMOS-based SPAD array detector (PRISM Light detector LVDS outputs, Genoa Instruments). For the experiments of this work, we registered only the $5 \times 5$ central elements of the $7 \times 7$ SPAD array. Finally, we integrated the BrightEyes-TTM into a Nikon AXR confocal microscope system (Supplementary Fig. S13a). The microscope was equipped with a 485 nm diode pulsed laser (ISS) and a conventional GaAs PMT (H7422P, Hamamatsu) to implement conventional FLIM. A constant-fraction discriminator (CFD, Flim Labs) was used to convert the analog PMT output signal to the digital input signal requested by the BrightEyes-TTM.

**Control system.** To control the custom laser-scanning microscopes, we built a LabVIEW-based system inspired by the Carma microscope control system[16,41]. The LabVIEW control system uses an FPGA-based general-purpose National Instruments (NI) data-acquisition-card (USB-7856R; National Instruments) to control all microscope components, such as the galvanometric mirrors, the piezo stage, and to initialize the SPAD array detector. The control system delivers the pixel, line, and frame clocks as digital TTL outputs. Finally, the control system is able to count the digital TTL photon signals from the SPAD array detector and transfer the data to the PC (via USB 2.0), where a LabVIEW-based software allowed real-time visualization of the images—in the case of imaging, or the intensity time traces and correlations—in the case of FFS. When we use the custom laser-scanning microscopes in the time-tagging modality, we switch the photon signals from the NI cards to the BrightEyes-TTM. In this case, the BrightEyes-TTM reads the photon signal from the SPAD array detector thanks to a custom I/Os daughter card connected via the FPGA mezzanine connector (FMC), Supplementary Fig. S21. Notably, we used different cards to match the detector cable connector and the signals standard, e.g., TTL or LVDS. The same I/Os daughter cards were used to transmit the duplicate signal from the central element of the SPAD array to the LabVIEW control system. The pixel, line, and frame signal passed through a custom buffer, to match the impedance between the NI card and the

Xilinx card, before being connected to an SMA digital input. The SYNC signal provided by the laser driver was converted from nuclear-instrumentation-module (NIM) to TTL (NIM2TTL Converter, Micro Photon Devices) and read by the Xilinx card via an SMA digital input. The data recorded by the BrightEyes-TTM was transferred to a PC via USB 3.0. To compare the BrightEyes-TTM with a commercial reference TTM, the SYNC (NIM) signal and the signal from the central element of the SPAD array could also be sent to a commercial multi-channel TTM (DPC-230, Becker & Hickl GmbH), running on a dedicated PC. For the experiments with the Nikon AXR confocal system, we control the microscope with the NIS elements software and we obtained the pixel, line, and frames clock from the Nikon control unit. Also in this case, we used a buffer to match the impedance between the control unit and the Xilinx card. The laser driver already provides a TTL SYNC signal.

### Data transfer

**The data structure.** To transfer the data from the TTM to the PC, we designed a simple data protocol, whose major advantages are the scalability to add photon channels and its flexibility. Briefly, the protocol perceives the SP detector array as a fast camera with a maximum frame rate of 240 MHz, i.e., the frequency at which the whole TTM architecture works. Under this scenario, the data protocol foresees a frame-like data structure streamed to the communication port (USB 3.0 in this implementation) in 32-bits long words, Supplementary Fig. S24.

The data are transmitted in a data structure composed of 16-bit words: each word contains a 7-bit identifier (ID), a 1-bit valid flag, and an 8-bit payload. The data structure contains a header (5 words) and the channel data. The number of words in the channel data is not fixed. In typical applications, the expected number of channels hit by a photon within the 240 MHz frame-rate window is very low. Therefore, a zero-suppression algorithm was implemented that prevents the transmission of data from no-hit channels.

The channel data words (ID < No. channels) contain for each channel: (i) 8 bits representing the value measured by the tapped-delay line of the respective photon channel $\Delta T_{\mathrm{START}}(ch)$; (ii) the channel data-valid ($V_n$) boolean-flag which confirms that an event in that channel has occurred.

The header (ID > 122) includes the following information: (i) 3 bits used as boolean-flag for the reference (REF) events (in our applications the pixel, line, and frame clocks); (ii) 8 bits representing the value measured by the tapped-delay line of the SYNC channel $\Delta T_{\mathrm{STOP}}$ (in our application the synchronization signal from the pulsed laser); (iii) the "laser data valid" (VL) boolean-flag which confirms that a SYNC event has occurred; (iv) 16 bits representing the number of clock cycles (STEP) of the free-running 240 MHz counter; (v) 8 spare bits that can be used for debugging purposes.

To reduce the data throughput, an event is transmitted only if one of the following conditions occurs: (i) a START event in one of the photon channels; (ii) a STOP event following a START event, e.g., a laser SYNC event occurs after a photon event (event filter); (iii) a REF event, e.g., a pixel/line/frame event; (iv) a force-write event. The force-write event is fundamental for reconstructing any time measurement (relative or absolute) that requires the coarse counter values. Indeed, since the data structure uses only 16 bits to store the 240 MHz coarse counter value, the counter resets every 273 μs. To guarantee the possibility of always reconstructing the relative and absolute values for each event, we implemented an internal trigger that forces the transmission of the data structure at least once every 17 μs, which corresponds to one-sixteenth of the coarse counter reset period.

To improve the robustness of the data transfer process, we mitigated the data throughput peaks by buffering the data in a large (512 kB) BRAM-FIFO on the same FPGA. The TTM code contains a mechanism that guarantees that in case the FIFO is filled over a certain threshold, i.e., the average data throughput exceeds the USB 3.0 data

bandwidth (because of PC latency, a high rate of events, or other reasons), the TTM temporarily enters a fail-safe mode, giving priority to some type of events. For example, in the case of imaging, the TTM gives priority to the pixel/line/frame flags and to the force-write events. This strategy guarantees the reconstruction of the absolute time of each event (photon or REF) and the image.

In the current TTM implementation, a USB 3.0 bus transfers the data to the PC. The data receiver software checks the data integrity of the data structures received (i.e., the IDs must be in the correct order) and, if the data structure is properly received, it sequentially writes the data to a binary file without any processing. The data receiver uses the *libusb-1.0* and is developed in the C programming language, Supplementary Fig. S25.

**The data pre-processing.** To create a user-friendly dataset, we pre-processed the binary file, Supplementary Fig. S26. First, the binary data is read as a table, which is saved to HDF5 (raw data). Since the free-running counter has 16 bits, it resets every 273 μs. Therefore, to obtain a consistent monotonic counter $n$, we update the 16 bits long free-counter provided by the data structure: when the free-counter value is lower than the one in the previous data structure, the value $2^{16}$ is added to it and added to the following counter values.

With the monotonic counter $n$, it is possible to calculate the arrival time of each photon event with respect to a REF event or with respect to the SYNC event (start–stop time). While the former calculation is trivial, the latter is more complex. Since the TTM architecture uses a start–stop reverse strategy to reconstruct the photon start–stop time, it is necessary to collect information about successive SYNC laser events. This information can be contained in the same data structure as the photon, or in a successive data structure. The pre-processing step identifies for each STOP event the corresponding START event and creates a table. Each table row contains a STOP event and includes (i) the relative monotonic counter $n_{\mathrm{SYNC}}$, (ii) the relative TDL value $\Delta T_{\mathrm{STOP}}$; an entry for each photon event linked to this specific STOP event. In particular, each entry contains: (i) the number of elapsed clock cycles $\Delta n(ch) = n_{\mathrm{SYNC}} - n_{\mathrm{photon}}(ch)$ and (ii) the TDL value $\Delta T_{\mathrm{START}}(ch)$.

To simplify the reconstruction of the image, we also pre-processed the REF information for the pixels, lines, and frames. We used the pixel/line/frame events to include in the table the columns $x$, $y$, and $fr$. The pixel event increases the $x$ counter; in the case of a line event, the $x$ counter resets and the line counter $y$ increases; in the case of a frame event, both the $x$, $y$ counters reset and the $fr$ counter increases.

We saved the pre-processed dataset in a single (optionally compressed) HDF5 file composed of different tables: one main table and 25 other tables referred to as the photon channels. All tables use a unique column identifier (idx) which allows the application software (e.g., FLISM software, FLFS software) to easily merge the information. The main table has a row for each SYNC channel event. Each row contains the corrected monotonic counter $n_{\mathrm{SYNC}}$, the coordinates $x$, $y$, $fr$, the TDL value $\Delta T_{\mathrm{STOP}}$ value, and the unique row index idx. The photon channel tables have a row for each photon. Each row contains the TDL value $\Delta T_{\mathrm{START}(ch)}$, the elapsed clock cycle $\Delta n(ch)$, and the index idx of the row of the corresponding sync event.

**The data calibration.** The HDF5 file contains all the information received by the TTM, but the data are structured in a way that is easier to access. However, this information still contains numbers of cycles or tapped delays. An off-line calibration phase allows transforming this information into temporal information. In particular, the calibration (Supplementary Note 1 and Supplementary Fig. S27), transforms each TDL value $\Delta T_{\mathrm{START}}(ch)$ or $\Delta T_{\mathrm{SYNC}}$ into a temporal value $\Delta t_{\mathrm{START}}(ch)$ or $\Delta t_{\mathrm{SYNC}}$, which we can use to calculate all (relative and absolute) temporal signatures of each event. The output of the calibration is again an

HDF5 file with a structure similar to the uncalibrated file. The main table has a row for each SYNC channel event. Each row contains the absolute SYNC time $t_{SYNC} = n_{SYNC} \cdot \mathcal{T}_{sysclk} + \Delta t_{SYNC}$, the coordinates $x, y, fr$ and a unique row index idx. The photon channel tables have for each row the start–stop time $\Delta t(ch)$, and the index idx of the row of the corresponding SYNC event.

**The application-dependent analysis.** Depending on the application, we further processed the calibrated HDF5. In general, any application requires the generation of the start–stop time (or TCSPC) histogram, which is simply the histogram of a series of $\Delta t$ values. Because of the autocalibration steps (Supplementary Note 1), the $\Delta t$ values are float values, thus the bin width of the histogram (i.e., the temporal resolution) can be chosen almost arbitrarily by the user (Supplementary Fig. S28). In this work, we used 48 ps, which is a value well below the IRF of the systems used in this work (e.g., 200 ps, Supplementary Fig. 1) and in the range of the average delay values (Supplementary Fig. S29) of the FPGA delay element forming the tapped-delay lines (Supplementary Note 1). In the case of FLISM analysis, the calibrated data are binned into a multidimensional photon-counts array $(ch, x, y, fr, \Delta t)$. The $\Delta t$ dimension is the start–stop histogram for a given set of $(ch, x, y, fr)$ coordinates. This multidimensional array can be saved in HDF5 and further processed with ad hoc scripts or software such as ImageJ. Importantly, because each SPAD element and each TDC can introduce a different, but fixed, delay, it is necessary to temporally align the histograms of the different channels: We performed a reference measurement under identical conditions with freely diffusing fluorescein. The resulting histograms were plotted in the phasor space. The phase difference between the expected position in the phasor space (given the known lifetime of 4.1 ns of fluorescein) and the measured position for each channel was used to calibrate each channel, i.e., to shift the histograms of the actual measurement back to the correct position. In the case of FLFS analysis, we ignored the $x$, $y$, and $fr$ information of the calibrated HDF5 and did not apply any further processing. Indeed, the FLFS needs a list of photon events for each channel, in which each photon has an absolute time—with respect to the beginning of the experiment, and the start–stop time ($\Delta t$). By using $t_{SYNC}$ as absolute time (instead of $t_{photon}$), the calibrated HDF5 file contains already all information structured in the best way.

## FLISM and FLFS analysis

**Image reconstruction and analysis.** We reconstructed the ISM images with the adaptive pixel-reassignment method[16]. In short, we integrated the 4D dataset $(ch, x, y, \Delta t)$ along the $\Delta t$ dimension, we applied a phase-correlation registration algorithm to align all the images $(x, y|ch)$ with respect to central image. The registration generated the so-called shift-vector fingerprint $(s_x(ch), s_y(ch))$. To obtain the ISM intensity-based images, we integrated the shifted dataset along the $ch$ dimension. To obtain the lifetime-based ISM image, we started from the 4D dataset (ch, $x, y, \Delta t$), Supplementary Fig. S30; for each $\Delta t$ value, we used the same shift-vector fingerprint to shift the relative 2D image; we integrated the result along the $ch$ dimension; we used the resulting 3D dataset $(x, y, \Delta t)$ and the FLIMJ software[42] to obtain the $\tau_{fl}$ maps (fitted with a single-exponential decay model). Alternatively, we applied the phasor analysis on the same 3D dataset. We calculated the phasor coordinates $(g, s)$ using cosine and sine summations[20,43]. To avoid artefacts, we performed the MOD mathematical operation of the TCSPC histograms with the laser repetition period value[43]. To calibrate the acquisition system and thus account for the instrument response function of the complete setup (microscope, detector, and TTM), measurements were referenced to a solution of fluorescein in water.

To demonstrate the spatial resolution enhancement achieved by ISM, we performed a Fourier ring correlation (FRC) analysis[44,45]. The FRC analysis requires two near-identical images obtained from two different measurements of the same sample in order to contain different noise realizations. The two images are correlated to obtain the effective cut-off-frequency (i.e., the frequency of the specimen above the noise level) of the images, and thus the effective resolution. Since, in this work, we built up the images photon-by-photon, we used the temporal tags of the photons to generate simultaneously two 4D data-sets $(ch, x, y, \Delta t)$. Then, we used the two data-sets to reconstruct the two statistically independent ISM images for the FRC analysis. In particular, we odd-even assign each photon to one of the two images by using the $\Delta T_{START}$ integer values. As explained for the sliding-scale approach, the photons are distributed uniformly across the START and STOP tapped-delay lines; thus, the method generates two statistically independent data-sets with similar photon counts.

**Fluorescence correlation calculation and analysis.** We calculated the correlations directly on the lists of absolute photon times[46]. For the sum 3 × 3 and sum 5 × 5 analysis, the lists of all relevant SPAD channels were merged and sorted. The data was then split into chunks of 10 or 5 s, for the beads and eGFP in living cells, respectively, and for each chunk, the correlation was calculated. The individual correlation curves were visually inspected and all curves without artifacts were averaged. To obtain the filtered correlation curves for the beads sample, the same procedure was followed, except that a weight was assigned to each photon based on its start–stop time[28,29]. The weights were obtained from the start–stop time histograms of each channel. Background counts coming from other sources than fluorescence, such as dark counts and detector afterpulsing, appear as an offset in the histogram because of their uniform probability distribution on the time scale of the histogram (25 ns). As a result, bins in which the number of photons approaches this offset value can be considered to contain only background. Having access to the start–stop time with ps range resolution, the BrightEyes-TTM allows classifying every photon in either of two classes: (i) almost certainly background or (ii) possibly fluorescence. In the former case, the photon can be completely removed from the data; in the latter case, the filter function is used to add a weight to the photon depending on how likely it is that the photon comes from the sample. Here, only counts in time bins between the peak of the histogram and about 10 ns later were included. The cropped TCSPC histogram of each channel was fitted with a single exponential decay function $H(t) = A \exp(-t/\tau_{fl}) + B$, with amplitude $A$, lifetime $\tau_{fl}$, and offset $B$ as free fit parameters. The filters were calculated assuming a single exponential decay with amplitude 1 and lifetime $\tau_{fl}$ for the fluorescence histogram and a uniform distribution with value $B/A$ for the background component. Then, for each count, a weight was assigned equal to the value of the fluorescence filter function at the corresponding start–stop time; counts that were detected directly after the laser pulse were assigned a higher weight than counts detected some time later, since the probability of a count being a fluorescence photon decreases with increasing start–stop time. The second filter function describes the background component and was not used for further analysis.

For both single-point and circular scanning[47] FCS, the correlation curves were fitted with a 1-component model assuming a Gaussian detection volume. For the circular FCS measurements[47], the periodicity and radius of the scan movement were kept fixed while the amplitude, diffusion time, and focal spot size were fitted. This procedure was used for the fluorescent beads and allowed calibrating the different focal spot sizes (i.e., central, sum 3 × 3 and sum 5 × 5). For the conventional FCS measurements, the focal spot size was kept fixed at the values found with circular scanning FCS, and the amplitude and diffusion time were fitted. Since we approximated the PSF as a 3D Gaussian function with a $1/e^2$ lateral radius of $\omega_0$ and a $1/e^2$ height of $k \times \omega_0$ (with $k$ the eccentricity of the detection volume, $k = 4.5$ for the central element probing volume, $k = 4.1$ for the sum 3 × 3 and sum 5 × 5 probing volume), the diffusion coefficient $D$ can be calculated from the diffusion time $\tau_D$ and the focal spot size $\omega_0$ via $D = \omega_0^2/(4\tau_D)$. All analysis

scripts are available in our repository and are based on Python (Python Software Foundation, (day 1) Python Language Reference, version 3.70. Available at http://www.python.org).

## Samples preparation

**BrightEyes-TTM characterization.** For the characterization and validation of the TTM, we used fluorescein (46955, free acid, Sigma-Aldrich, Steinheim, Germany) and potassium iodide (KI) (60399-100G-F, BioUltra, ≥99.5% (AT), Sigma-Aldrich). We dissolved fluorescein from powder into DMSO (Sigma-Aldrich) and then we further diluted it to a 1:1000 v/concentration by adding ultrapure water. For the fluorescein quenching experiments, we diluted the 1:1000 fluorescein solution at different volume ratios with the KI quencher (from 1:2 to 1:256). All samples were made at room temperature. A fresh sample solution was prepared prior to each measurement.

**Imaging.** For the imaging experiments, we used (i) a sample of 100 nm fluorescent beads (yellow-green FluoSpheres Q7 Carboxylate-Modified Microspheres, F8803; Invitrogen). We treated the glass coverslips with poly-L-lysine (0.1% (w/v) in $H_2O$, P8920, Sigma-Aldrich) for 20 min at room temperature, and then we diluted the beads in Milli-Q water by 1:10,000 v/v. We drop-casted the beads on the coverslip and washed the coverslip after 10 min with ultrapure water. Then, the glass coverslip was dried under nitrogen flow and mounted overnight with Invitrogen ProLong Diamond Antifade Mounting Medium (P36965); (ii) Hippocampal mice neurons expressing super-ecliptic-pHluorin (SEP)-tagged-$\beta$3 subunit $\gamma$-aminobutyric acid, type A (GABAA) receptor for live-cell imaging experiments. Hippocampal neuronal cells were isolated and dissected from early postnatal (day 1) B6;129-Nlgn3wt/J (B6129SF1/J) mice of either sex using a previously published protocol[48], in accordance with the guidelines established by the European Communities Council (Directive 2010/63/EU of 22 September 2010) and by the national legislation (D.Lgs.26/2014). After the dissection, hippocampal neurons were plated onto glass coverslips coated with poly-D-lysine (0.1 µg/ml, Sigma-Aldrich) and maintained in Neurobasal-A medium supplemented with 1% GlutaMAX™, 2% B-27 (all from ThermoFisher Scientific, Italy) and 5 µg/ml Gentamycin at 37 °C in 5% $CO_2$. After 8 days in vitro (DIV8), neurons were transfected with SEP-tagged-$\beta$3 subunit GABAA receptor[49] using Effectene Transfection Reagent (Qiagen, Italy) according to the manufacturer's recommendations. Measurements were made at DIV14 in Live-Cell Imaging Solution (ThermoFisher Scientific) at 37 °C. (iii) a microscope slide containing fixed HeLa cells stained for $\alpha$-tubulin. We cultured HeLa cells in Dulbecco's Modified Eagle Medium (DMEM, Gibco, ThermoFisher Scientific) supplemented with 10% fetal bovine serum (Sigma-Aldrich) and 1% penicillin/streptomycin (Sigma-Aldrich) at 37 °C in 5% $CO_2$. The day before immunostaining, we seeded HeLa cells on coverslips in a 12-well plate (Corning Inc., Corning, NY). The day after, we incubated cells in a solution of 0.3% Triton X-100 (Sigma-Aldrich) and 0.1% glutaraldehyde (Sigma-Aldrich) in BRB80 buffer (80 mM Pipes, 1 mM EGTA, 4 mM MgCl, pH 6.8, Sigma-Aldrich) for 1 min. We fixed HeLa cells with a solution of 4% paraformaldehyde (Sigma-Aldrich) and 4% sucrose (Sigma-Aldrich) in the BRB80 buffer for 10 min, and then we washed them three times for 15 min in phosphate-buffered saline (PBS, Gibco™, ThermoFisher Scientific). Next, we treated the cells with a 0.25% Triton-X-100 solution in BRB80 buffer for 10 min and washed three times for 15 min in PBS. After 1 h in blocking buffer (3% bovine serum albumin (BSA, Sigma-Aldrich) in BRB80 buffer), we incubated the cells with monoclonal mouse anti-$\alpha$-tubulin antibody (1:1000, Sigma-Aldrich) diluted in the blocking buffer for 1 h at room temperature. The alpha-tubulin goat anti-mouse antibody was revealed by Alexa Fluor 488 goat anti-mouse (1:1000, Invitrogen, ThermoFisher Scientific) incubated for 1 h in BRB80 buffer. We rinsed HeLa cells three times in PBS for 15 min. Finally, we mounted the coverslips onto microscope slides (Avantor, VWR International) with ProLong

Diamond Antifade Mountant (Invitrogen, ThermoFisher Scientific). (iv) Live-cell imaging of HeLa cells stained for di-4-ANEPPDHQ. The day before live-cell imaging, we seeded HeLa cells in a µ-Slide eight-well plate (Ibidi, Grafelfing, Germany). Just before measurement, cells were incubated with DMEM (Gibco, ThermoFisher Scientific) supplemented with 5 µM di-4-ANEPPDHQ (Invitrogen, ThermoFisher Scientific) at 37 °C in 5% $CO_2$ for 30 min. The cells were washed three times with DMEM. Measurements were made in Live-Cell Imaging Solution (ThermoFisher Scientific) at 37 °C.

**FLFS experiments.** For fluorescence fluctuation spectroscopy experiments, we used: (i) YG carboxylate fluoSpheres (REF F8787, 2% solids, 20 nm diameter, actual size 27 nm, exc./em. 505/515 nm, Invitrogen, ThermoFisher) diluted 5000× in ultrapure water. A droplet was poured on a coverslip for the FLFS measurements; (ii) goat anti-Mouse IgG secondary antibody with Alexa Fluor 488 sample (REF A11029, Invitrogen, ThermoFisher). The antibody was diluted 100× in PBS to a final concentration of 20 µg/mL. In total, 200 µL of the resulting dilution was poured into an eight-well chamber previously treated with a 1% BSA (ThermoFisher Scientific) solution to prevent the sample from sticking to the glass. All samples were prepared at room temperature. A fresh sample solution was prepared for each measurement. (iii) HEK293T cells expressing monomeric-eGFP. HEK293T cells (Sigma-Aldrich, non-authenticated cell line) were cultured in DMEM (Dulbecco's Modified Eagle Medium, Gibco™, ThermoFisher Scientific) supplemented with 1% MEM (Eagle's minimum essential medium), Non-essential Amino Acid Solution (Sigma-Aldrich), 10% fetal bovine serum (Sia-Aldrich) and 1% penicillin/streptomycin (Sigma-Aldrich) at 37 °C in 5% $CO_2$. HEK293T cells were seeded onto a µ-Slide eight-well plate (Ibidi GmbH). HEK293T cells were transfected with pcDNA3.1(+)eGFP (Addgene plasmid #129020). Transfection was performed using Effectene® Transfection Reagent (Qiagen, Hilden, Germany) according to the manufacturer's protocol. Measurements were performed in Live-Cell Imaging Solution (ThermoFisher Scientific) at room temperature. As the experiment performed with HEK cell is not related to relevant biological processes specific for this cell line, its authentication is, in this case, not relevant.

## Ethical approval declarations

The experiments involving primary neuronal cultures from mice were prepared in accordance with the guidelines established by the European Communities Council (Directive 2010/63/EU of 22 September 2010) and following the Italian law D.Lgs.26/2014. All the animal procedures have been approved by the Italian Ministry of Health Regulation (Authorization 800/2021-PR) and by the Italian Institute of Technology welfare body.

## Reporting summary

Further information on research design is available in the Nature Portfolio Reporting Summary linked to this article.

# Data availability

The raw time-tagged data generated in this study have been deposited in Zenodo under the accession code https://doi.org/10.5281/zenodo.4912656. Full build instructions for the BrightEyes-TTM have been deposited in GitHub and ReadTheDocs under accession codes https://github.com/VicidominiLab/BrightEyes-TTM and https://brighteyes-ttm.readthedocs.io.

# Code availability

The firmware and the VHDL/Verilog source code for implementing time-tagging on the FPGA evaluation board, the data receiver software to install on the personal computer, and the operating software have been deposited in GitHub under accession code https://github.com/VicidominiLab/BrightEyes-TTM and Zenodo[50,51].

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

## Acknowledgements

This research was supported by Fondazione San Paolo, "Observation of biomolecular processes in live-cell with nanocamera", No. EPFD0098 (E.S. and G.V.), by the European Research Council, Bright Eyes, No. 818699 (G.T. and G.V.), and by the European Union's Horizon 2020 research and innovation programme under the Marie Sk1odowska-Curie grant agreement no. 890923 (SMSPAD) (E.S. and G.V.). We thank Prof. Alberto Diaspro and Dr. Paolo Bianchini (Nanoscopy & NIC@IIT, Istituto Italiano di Tecnologia) for useful discussions; Dr. Michele Oneto (Nikon Imaging Center) and Marco Scotto (Molecular Microscopy and Spectroscopy, Istituto Italiano di Tecnologia) for support on the experiments; Alessandro Barcellona (Electronic Design Laboratory, Istituto Italiano di Tecnologia) for design and implementation of the custom-made buffer; Prof. Alberto Tosi, Prof. Federica Villa, Dr. Mauro Buttafava (Politecnico di Milano), Dr. Marco Castello, and Dr. Simonluca Piazza (Istituto Italiano di Tecnologia and Genoa Instruments) for useful initial discussions in the time-to-digital design and for the realization of the single-photon-avalanche-diode detector array; All members of the RNA Initiative at the Istituto Italiano di Tecnologia for their contribution to the long-term vision of this project.

## Competing interests

G.V. has a personal financial interest (co-founder) in Genoa Instruments, Italy; A.R. has a personal financial interest (founder) in FLIM LABS, Italy, outside the scope of this work. The remaining authors declare no competing interests.
