## [Peer Review File · Nature Communications]

The BrightEyes-TTM as an Open-Source Time-Tagging Module for Democratising Single-Photon MicroscopyREVIEWER COMMENTS

Reviewer #1 (Remarks to the Author):

COMMENTS for the AUTHORS:

The paper provides a very detailed and nicely written overview of an open-source time-tagging module (TTM) designed for use in combination with emerging single-photon avalanche diode (SPAD) small array photodetectors, to be used in single-photon laser-scanning microscopy. The developments leverage on the advances in field-programmable gate array technology (FPGA), which allow the implementation of hardware functionalities such as (multi-channel) time-to-digital converters, with the potential for reconfigurability. Several application examples are highlighted, such as FLISM and FLFS.

The real added value of the paper is in the fully open source implementation, tailored for SPAD-based LSM, extensive characterisation, and system integration aspects. Commercial FPGA-based multi-channel time-tagging modules do exist, but are still pricey; from the user perspective, an open source implementation is therefore certainly welcome. Time will tell how the microscope and/or add-on manufacturers react (the Authors used a custom-built microscope), also given that SPAD arrays like the one used are just emerging and being made available to the community.

The main issues to be addressed and discussed before publication include:

- **Scaling to other platforms, reproducibility, possible increase of timing resolution for interfacing to next-generation SPADs, end user support.**
- **Your vision on the availability of SPAD detectors – also other models – and how easy it is/would be to interface them to the proposed platform on one side and existing microscopes on the other.**
- **Comparison with TDC state-of-the-art, e.g. in form of a table, ideally both research and commercial.**
- **Calibration (p. 11): how often is a calibration necessary, how easy is it? Are there drift issues, e.g. with temperature changes, in particular for longer experiments, or as a function of the photon rates, and therefore TDC activity?**
- **Taking one step back: the Authors have made the basic assumption that SPAD arrays are the way to go, which I definitely share. However, they might need to better motivate their case.**

DETAILED COMMENTS:

- **Main section (pp. 1-2): an introductory/overview image, e.g. featuring a general microscopy and detector set-up, could be really useful for the general reader, as well as a couple of lines explaining the SPAD fundamentals and related state-of-the-art.**
- **Results:**
 - **It might be useful to add, already at this level, a couple of sentences on the key TDC general aspects, and how their specifications were chosen as a function of the target SPAD array and/or application. (A description comes later, p. 8.)**
 - **Maybe add one line highlighting the TDC results quality (e.g. excellent DNL/INL)**
 - **Fig. 1f: if appropriate add a short comment on the points which fall outside of the universal semi-circle.**

- o **Fig. 2:** I would suggest to add a label on top of each column to make it easy to understand that the left one refers to "closed confocal", etc., without the need to read the full caption, and possibly some other keywords if appropriate.
- o **Fig. 2:** which is the acquisition time? Is it possible to draw a comparison with what a "standard" AiryScan module would provide?
- o **Is Fig. 2b** clearly referenced in the text?
- o **P. 4, uncorrelated background shifts:** although it's not the main point, is it not possible to correct for this via background subtraction?
- o **P. 5:** where is Table 3?
- **Methods:**
- o **Fig. 4:** the colours in the bottom left panel (light green and blue) are too similar.
- o **P. 9:** the total component cost is likely without the SPAD sensor itself, correct? Are there any additional costs due to licenses, e.g. software or firmware (such as proprietary IP blocks or libraries)?
- o **P. 10: custom PCBs:** do they represent a potential obstacle for the open-source approach? Does every end user need have them manufactured by him/herself?
- o **Pp. 9-11:** this long section is really a bit hard to follow without often resorting to the supporting images in the SuppMat -> can the situation be improved? Maybe by adding a "summary" image?
- o **P. 12:** "In this way, photons that are likely to not originate from the sample were removed." -> can this be clarified?
- o **P. 12: diffusion times and related results:** do they match the state-of-the-art and/or your expectations?
- o **P. 12: "afterpulse component":** how much is afterpulsing (and the other key SPAD parameters)?
- **Bibliography:**
- o Some caps are missing, e.g. raman -> Raman, sted -> STED, etc
- **Bibliography - possible additional relevant literature:**
- o Recent review on ASIC TDC implementations for biophotonics applications: R. Scott, W. Jiang, and M. Jamal Deen, "CMOS Time-to-Digital Converters for Biomedical Imaging Applications", IEEE Review Biomedical Engineering, 10.1109/RBME.2021.3092197
- o Q-ISM SPAD-based implementation: G. Lubin, R. Tenne, I. M. Antolovic, E. Charbon, C. Bruschini, and D. Oron, "Quantum correlation measurement with single photon avalanche diode arrays," Optics Express, vol. 27, no. 23, Art. no. 23, Oct. 2019, doi: 10.1364/OE.27.032863.
- o Other FPGA-based multi-channel TDC implementation: S. Burri, C. Bruschini, and E. Charbon, "LinoSPAD: A Compact Linear SPAD Camera System with 64 FPGA-Based TDC Modules for Versatile 50 ps Resolution Time-Resolved Imaging," Instruments, vol. 1, no. 1, Art. no. 1, 2017, doi: 10.3390/instruments1010006.

• **Supplementary:**

o **P. 16:** bin width: how small could you go in the case of (future generation) SPADs with much improved temporal resolution? Would you still be limited by design to a precision of ~ 30 ps?

o **P. 17:** which is the relationship between the histogram bin width of 48 ± 3 ps, and the previously reported 48 ± 25 ps?

o **P. 18:** Fig. S1b: is the linearity unit (i.e. arbitrary units) correct?

o **P. 39:** Fig. S22: how are the outliers taken into account?

TYPOS (non-exhaustive list):

(Fig. 1) pane -> panel?, (p. 7) "version able to handle up to 49 photon channels Whilst" -> full stop missing, "course counter" -> coarse, (p. 8 right 4th line from the bottom) "t_sysclk" or "T_sysclk"?, (p. 9) "temporally correlated" or uncorrelated?, "tap-delay line" -> tapped-delay line? (or tapped delay line?) , "the photons-signals", "Since the TTM architecture uses a start-stop reverse strategy to reconstruct the photon start-stop time – for a given channel ch, it is..." -> unconventional writing, maybe rephrase, CARR4 -> CARRY4, "The inversion of these values" -> reciprocal?, "recording the simultaneously the signal", "fingerprint ans summed" -> and.

Reviewer #2 (Remarks to the Author):

Rosetta et al. report on the development of an open-sources multi-channel time-tagging module designed to implement and democratise current and future fluorescence single photon laser scanning microscopy (SP-LSM) techniques. They claim that their FPGA-based approach enables quick prototyping, updating and adaptation. The module was integrated in an existing confocal laser scanning microscope equipped with a 5x5 SPAD array to perform fluorescence lifetime image scanning microscopy (FLISM). The authors also claim that for the first time they demonstrate the combination of comprehensive correlation analysis (CCA) with fluorescence lifetime analysis using the diffusion mode of eGFP with its lifetime. They provide detailed guidelines, hardware parts lists and open-source code for FPGA firmware and operational software along with their manuscript.

Overall, I find this manuscript quite interesting and possibly suited for publication in Nat. Commun. after some major revisions as detailed below.

Major comments

1) p4, right column, 1st paragraph: The authors explain the deviation from the expected universal semicircle in the phasor plot by uncorrelated background. While this might be the most suitable explanation, it is not clear if it is the only explanation. Could this be proven e.g., by comparison with a different dye?

2) On p5, the authors discuss state the use of the phasor-based segmentation. What motivates the use of phasor-based segmentation? Can the authors please discuss possible advantages/disadvantages over the lifetime threshold-based approaches?

3) The authors mention different measurements using the B&H TCSPC system. However, from the shown data the reader can't decide what they will gain/lose when switching from one system to the other. 4) Can the authors please provide a clear concise benchmark comparing BrightEyes with, for example, the B&H TCSPC system?

5) Furthermore, I would be interested in the experimental saturation limit of the systems? Is it completely independent from label density/brightness?

6) The data shown in fig 2 compares in vitro data with data in fixed cells. It is not clear from the text if the differences observed for the lifetime histograms are due to different fluorescent labels or different chemical environments affecting the same fluorophore.

Furthermore it is not clear if there is a quality difference between the data from the phantom and the cell experiments. How does sample preparation affect the results? Is there anything interested labs will have to account for when choosing the BrightEyes approach?

7) Fig 5f shows different measurements (grey lines). However, the associated mean seems to have a significant bias (shifted downwards in resp. of the lifetime axis). Can the authors please explain/consolidate this?

8) In the discussion of the results shown in fig 5, readers might find it difficult to understand the relationship between mobility and the diffusion coefficients at different scales. I strongly recommend a clearer explanation for broader audience in Nat. Commun.

9) As the authors claim easy adaptation for future systems, they should include the limitations of their approach when it comes to larger SPAD arrays. Will this architecture allow to deal with SPAD cameras? What will be required to do this and could their approach be extended in that direction?

10) The authors claim that their system enables quick prototyping, updating and adaptation. However, on p 10 they mention a LabView control system that seem not be documented in their data.

11) On p10 the authors describe a fail-safe mode. However, it remains unclear to me if/how this is reflected in the data, so that the user can decide on how to proceed with it.

Minor comments

12) Can the authors explain the weak tendency for longer lifetimes in fig. 1e as compared to B&H (before B&H signal has cut-off)?

13) Are the arrows in fig 1d supposed to indicate FWHM (don't hope so)?

14) Can the authors explain the clear deviation of the INL from an equal distribution in fig 1a?

15) p4, left column, first paragraph: Can the authors please indicate in fig 1d the "additional bump" they discuss in the explanation?

16) In the discussion of fig 4c (p5) the authors state a zero intercept of the lifetime dependency. However, fig 4c shows an intercept close to zero. Can the authors please account for this deviation?

17) Fig S1d, am I mistaken or does the calculate standard deviation show weak correlations? Can the authors please look into this and explain?

18) Can the authors please briefly explain why in fig S3 the resolution in cells is smaller than in the phantom sample?

potential typos:

19) Fig 1 – figure caption, line 3: "differential non--linearity"

p2, right column, last paragraph: "(the 4 corner element are neglected)"

20) p7, right column, 1st paragraph: "... handle up to 49 photon channels Whilst, time-tagging module ..."

21) p10, right column, 2nd paragraph: the last sentence seems to be incorrect.

Reviewer #3 (Remarks to the Author):

Rosetta et al. present a technical tour-de-force on a homebuilt multichannel time tagging module called BrightEyes-TTM. They provide extensive characterization and a few application examples using a SPAD array. The methods and supplemental information are thorough. I would feel comfortable building a BrightEyes-TTM unit if we owned a SPAD array.

Before I can provide a final recommendation, I suggest that the authors streamline the manuscript to give the reader a clearer picture of the accomplishments. For example, is the focus the technical innovation, the experiments that BrightEyes-TTM enables, or both? Further controls and modeling may be required if the focus is to present new experiments. There also needs to be a more straightforward presentation of the

technical points in both the writing and figures.

Here I provide a set of major and minor questions/notes and suggest potential actions for each.

Major

1. Figure presentation.

All of the figures are complicated to follow. For example, Figure 4 contains a large amount of information presented on two different measurements, fluorescent beads, and Alexa488. In addition, the data is overwhelming, making it difficult to understand the results and how the BrightEyes-TTM architecture enabled the work.

Here I provide some suggestions for each figure, intending to improve readability:

Figure 1.

The cartoons in the top row are helpful to guide the reader. It would be helpful to provide a clear explanation of why each test is essential in the text. For example, the explanation on testing linearity is highly technical (page 3, first paragraph of left column). Why is this test important in the context of biological imaging? The same goes for all of the tests shown in Figure 1.

For a general audience journal, I suggest that some of the finer details of Figure 1 can be moved to the supplement and replaced with cartoon explanations in the main manuscript.

Figure 2.

The difference between the columns is not clear from looking at the figure. However, a simple cartoon showing the configuration would emphasize why the far left column has low signal-to-noise but similar shapes to the column to the right (0.2 AU vs. ISM).

The resolution increase is not apparent due to the lookup tables and saturation choices. Please present averaged bead profiles, with average FWHM and average FRC results on the vimentin in addition to cleaner intensity images.

Please explain how the rejection criteria for the lifetime histograms are determined. They appear aggressive for the vimentin. Why is this necessary?

Why are the lifetime distributions significantly different between the 0.2 AU and ISM/1.4 AU data? For the beads, this appears to be due to fewer photons. However, the distribution appears significantly broader for the vimentin with a shifted peak. Would the lifetime histograms match if the experiments collected the same number of photons?

How is Δt handled for the ISM data? It is not completely clear to me from the methods and supplemental material. Specifically, during photon reassignment, are there technical issues that must be accounted for to ensure the arrival time is correct?

Figure 3.

This is a great experiment. Unfortunately, the ISM images appear over-saturated, rendering the ISM contrast and resolution less impressive than it is. Please correct.

How is the segmentation performed on the phasor plot? Please describe the procedure in more detail. For example, how is the center and radius of the circles for each dye state determined? From another experiment or theory?

Because this figure focuses on live-cell data, it would be helpful to show dynamic information. For example, a supplemental movie showing the membrane dynamics over time using ISM and FLISM side by side would be a powerful visualization.

Figure 4.

Figure 4 has an overwhelming amount of information. A cartoon of circular scanning FLFS, why this removes the need for calibration, and why high background can influence the correlation function will help guide the reader.

The rationale behind the filtering used to account for detector afterpulsing, and background signal is unclear. Can the authors please expand on this in the methods and connect to their benchmarked performance in Figure 1?

Please see major point #2 below for questions on using the number of SPAD array pixels as a proxy for focal volume.

Figure 5.

This is another great experiment. I have a few questions about the results.

Please see major point #2 below for questions on using the number of SPAD array pixels as a proxy for focal volume.

It would be informative for the authors to provide the results for both the cytoplasm and nucleus.

Where is the center of the beam focus along the axial extent of the cell?

The cause of the variability in offsets and slopes for the dashed lines in (f) is not immediately clear. Can the authors please provide a clearer explanation of how the details of spot-variation analysis lead to the presented data?

Does the downward trend in $D_{\text{central}}/D_{5 \times 5}$ presented in panel (g) correspond with photobleaching? It does not appear so from the data in (c), I just wanted to confirm.

2. Relationship of excitation beam (true ω_0) and detector derived ω_0 .

In multiple areas of the manuscript, there are plots of data-derived quantities (concentration, diffusion constant, etc.) versus the excitation beam waist (ω_0) or the focal volume. The authors appear to calculate ω_0 based on the number of pixels on the SPAD array. How accurate is this? The intensity profile at the edges of the SPAD array may be significantly different from the center of the beam. Additionally, is the pixelation of the SPAD array taken into account when calculating the effective ω_0 ? Uncertainties in ω_0 will alter the downstream analyses.

There is a second issue, as the beam's intensity profile is not truly a 3D Gaussian. Therefore, how much correction is required when accurately modeling the focal volume, e.g., using a Gauss-Lorentz function (for example, see doi: 10.1016/S0006-3495(01)76264-9)?

3. Overall vision/impact

In their conclusion, the authors state:

"The principal aim of this work is to democratise single photon microscopy by giving to any microscopy laboratory the possibility to update their existing LSM systems. "

It is unclear to me how the BrightEyes-TTM helps update existing LSM systems without adding a SPAD array. All of the use cases in the paper are focused on SPAD array detectors. At this point, it is unclear how the community would adopt these methods without a SPAD array. I agree that the work presented here is an important step forward for the SPAD array community (and I wish we had a SPAD array to justify

building a BrightEyes-TTM!). However, as presented, the lack of applicability to standard laser scanning microscopes does diminish the impact of the author's statement of purpose.

Can the authors please elaborate on experiments or analyses that BrightEyes-TTM enables on a standard LSM that would not be possible otherwise?

Minor

1. When constructing the filtering functions for the correlation curves (page 12), the authors state:

"Only photons in time bins between the peak of the histogram and about 10 ns later were included. In this way, photons that are likely to not originate from the sample were removed. "

Could the authors please explain the source and number of the excluded photons? Are these actual photon events, detector noise, or electronics noise?

2. Shortly after the above statement, the authors further assert:

"Then, for each photon, a weight was assigned equal to the value of the fluorescence filter function at the corresponding photon start-stop time. E.g., photons that were detected directly after the laser pulse were assigned a higher weight than photons detected some time later, since the probability of a photon being a fluorescence photon decreases with increasing start-stop time."

Is this true in heterogeneous samples? The lifetime of both fluorescence dyes and proteins is highly sensitive to the local environment. I understand that having access to this data improves over prior capture cards. The increased data quality offers an opportunity to explain observations like this one.

3. Throughout the manuscript, there are many colloquial language usages, including more than one "In a nutshell,...". Please edit down and rewrite to communicate the essential points without such expressions.

--Douglas Shepherd

REVIEWER COMMENTS

Reviewer #1 (Remarks to the Author):

COMMENTS for the AUTHORS:

The paper provides a very detailed and nicely written overview of an open-source time-tagging module (TTM) designed for use in combination with emerging single-photon avalanche diode (SPAD) small array photodetectors to be used in single-photon laser-scanning microscopy. The developments leverage on the advances in field-programmable gate array technology (FPGA), which allow the implementation of hardware functionalities such as (multi-channel) time-to-digital converters, with the potential for reconfigurability. Several application examples are highlighted, such as FLISM and FLFS.

The real added value of the paper is in the fully open-source implementation, tailored for SPAD-based LSM, extensive characterization, and system integration aspects. Commercial FPGA-based multi-channel time-tagging modules do exist, but are still pricey; from the user perspective, an open-source implementation is therefore certainly welcome. Time will tell how the microscope and/or add-on manufacturers react (the Authors used a custom-built microscope), also given that SPAD arrays like the one used are just emerging and being made available to the community.

All authors thank the referee for the positive review and for appreciating the open-source aspect of the BrightEyes-TTM. The authors addressed all points raised by the referee – major changes in the manuscript are indicated in red – and think this substantially helped to improve the quality of the work.

A detailed description of all major and minor changes is written below in the point-to-point responses to the referee's questions, comments, and suggestions. However, before the point-to-point response, we would like to give the referee a general overview of how we approached the revision. From the beginning, the main scope of this work was to provide the "microscope makers" (or DIY microscopy community) with an open-source data-acquisition module to implement what we call single-photon microscopy. In short, a laser-scanning microscope able to register the fluorescence signal photon-by-photon and with a series of tags able to encode extra information from the sample. Within this aim, we implemented the BrightEyes time-tagging module (the BrightEyes-TTM), a data-acquisition system with characteristics matching this need. For example, although literature reports on different approaches – compatible with our FPGA technology – that can temporally tag photons with a precision of a few ps, achieving such precision was not our aim. On the other side, using relatively small (i.e., tens of pixels/elements) SPAD array detectors able to tag individual photons not only with temporal tags but also with spatial tags opens the need for a multi-channel time-tagging module. At the same time, as requested by the Editor in response to the manuscript pre-inquiry, we extensively validated the BrightEyes-TTM, not only on an already established single-photon microscopy technique, such as fluorescence lifetime image scanning microscopy (FLISM), but also on a brand-new technique – named fluorescence lifetime fluctuation spectroscopy (FLFS).

We reviewed the manuscript with these primary objectives in mind. On top of this, following the referee's suggestion, we further strengthened the open-source nature of the project: we demonstrated the scalability of the platform and its versatility in transforming existing laser-scanning microscopes into single-photon microscopes. Specifically, we:

- completely re-designed the data-transfer (to the computer) system and acquisition protocol to improve the scalability (in terms of channels) of the module. We used the new protocol to

increase the number of channels from 21 to 25 (a 49 channels version is also deployed, released in the GitHub repository, but not fully tested). Practically, the community may add extra channels by keeping in mind the limit of the FPGA resources and the USB 3.0 transmission band. In this context, we are exploring the possibility to connect the BrightEyes-TTM to the computer with a PCI-Express, thus improving the transmission band.

- validated the module with a commercial 7x7 SPAD array detector with LVDS photon output signals instead of TTL signals as for the SPAD array detector prototype used for the first submission, thereby demonstrating the ability of the module to adapt to different SPAD array detectors. This validation further opens to an effective dissemination of single-photon microscopy and, in general, pave the way to the use of small SPAD array detector in any laboratory. In this context, we made a new FMC daughter card to connect the new SPAD array detector to the BrightEyes-TTM. The file grabber of the new PCB is already publicly available.
- validated the module with conventional single-photon detectors, namely a commercial single-element SPAD detector, and a photomultiplier detector whose signal has been digitalized with a constant fraction discriminator. The authors are aware that to effectively create a large community of users for the BrightEyes-TTM and for the corresponding analysis software, also the laboratories in which a SPAD array detector is not available, can readily implement conventional FLIM or fluorescence-lifetime-correlation spectroscopy.
- used the module to transform a commercial laser-scanning microscope into a FLIM microscope. This demonstrates the versatility of the module to work also with commercial microscope control-units. The only requirement is that the control-unit provides as outputs the scanning synchronization signals.
- created a ReadTheDocs web page linked to the GitHub repository to host the documentation for the open-source project.

The main issues to be addressed and discussed before publication include:

- **Scaling to other platforms, reproducibility, possible increase of timing resolution for interfacing to next-generation SPADs, end user support.**

Being written in both HDL and Verilog, the BrightEyes-TTM project allows for easy porting and scaling to other Xilinx family 7 platforms or, more in general, to all Xilinx-based platforms that have the CARRY4 component available as primitive logic. CARRY4 is the small unit element upon which the whole TDC design is built.

Improving the timing resolution on the current architecture (i.e., time bin width or LSB dimension) is possible using all the CARRY4 sub-component capabilities (e.g., Tontini et al. IEEE Transactions on Nuclear Science, 65(2):680-690,2018), although the amelioration, for being really beneficial, should also affect all the other figures of merit of the overall TDC design such as DNL, INL, and single-shot precision. As of today, we envision that, in order to improve the timing resolution of the BrightEyes-TTM while keeping the other figures of merit consistent, a few adjustments to the source code (HDL/Verilog) may be required. In this context, we would like to stress more the spirit of the open-source project and the use of an FPGA development kit: the idea behind this is that “microscope makers” in the community will synergically drive and work on the future developments of this module by integrating well-established and new approaches to improve the time-tagging module characteristics. For example, the introduction of a new SPAD array detector into the market motivated us to increase the number of input channels in the module.

The BrightEyes-TTM can also be transposed and deployed, with minor code adjustments, in other and more performant Xilinx-based platform architectures like the ZYNQ, UltraScale and Ultrascale+ which, for example, instead of the CARRY4 logic have the CARRY8 primitive inbuilt. It is known from the literature that the CARRY8 logic yields better timing resolution compared to the CARRY4 but it remains still unknown how the use of CARRY8 affects the other figures of merit in the specific BrightEyes-TTM architecture.

Porting to other Xilinx FPGA platforms is possible and feasible but this option is not a turnkey solution as it requires HDL/Verilog code changes to use the CARRY8 components. In addition, all figures of merit would require a complete re-evaluation and assessment.

Moreover, it is already possible in the current architecture, with minor modifications, to improve the timing resolution using more than one tapped delay-line per single input channel. By averaging these signals, the timing resolution increases. The FPGA platform resources determine the maximum number of tapped delay-lines implementable in the design. The compromise between the number of channels and the timing-resolution can be defined during the code design depending on the application.

In conclusion, there is a lot of room to improve the time-resolution and the other characteristics of the BrightEyes-TTM, both by exploring the many architectural approaches reported in the literature (e.g., CARRY4/8 sub-sampling, multiple tapped delay-line, wave-union) or migrating to a new generation FPGA with improved technological nodes. However, the current BrightEyes-TTM characteristics are optimal for the single-photon microscopy application, and demonstrating the portability of the architecture is out-of-the-scope of this work.

We partially introduced the outcome of the above discussion in the Main Text by adding the following sentences:

“In conclusion, the BrightEyes-TTM offers a combination of single-shot precision, linearity, temporal range, number of channels, and sustained count rate which is suitable for measuring fluorescence lifetime with state-of-the-art SPAD array detectors for SP-LSM. It is worth noting that literature reports on different TDC implementations based on the same Kintex 7 FPGA family (Kintex 7) with superior characteristics (Supplementary Table S2). This aspect indicates the possibility of further improving the characteristics of our BrightEyes-TTM to match the expected enhancement in performances (e.g., reduction of time jitter) in the next SPAD array detector generations. In this context, it is important to observe that the current 25 channels TTM implementation (as well as the 49 channels implementation of the next BrightEyes-TTM release) uses a small portion of the FPGA resources available (Supplementary Fig S6), thus offering room for implementing strategies able to improve the overall characteristics of our TTM. For example, the single-shot precision of each channel can be readily improved by implementing a multichain measurement averaging method (21): The input signal of each channel is connected to multiple TDCs simultaneously, and the final time is obtained by averaging the single TDC values.”

“Whilst the time-tagging module implements the most informative DAQ system strategy, it requires transferring (to the PC) and storing a consistent amount of data. Thus, this approach is not practically scalable to the large (megapixel) SPAD cameras requested by wide-field microscopy (34). Here, typically an application-specific integrated circuit (ASIC) TDC is implemented directly on each SPAD element (35), and the temporal information is pre-processed to reduce the amount of data to transfer. For example, during the measurement, the detector records the start-stop histogram, rather than the photon time-tags. However, also in the context of small SPAD array detectors, it could be important to reduce the data transferred to the PC. For this reason, a potential direction for the next developments in the BrightEyes-

TTM project could be the migration to a new FPGA development kit equipped with ARM-based processors. In case of migration to another development kit, it may be also interesting to explore the new generation FPGA with improved technological node (e.g., Ultrascale 20-nm), on which TDCs with superior performances are demonstrated (36).

We are fully aware of the importance of documentation in an open-source project. For this reason, in this new reviewed version of the project, we provided further support to the end-user by implementing a ReadTheDocs page based on the GitHub repository. The combination of ReadTheDocs and GitHub allows the user to report issues, bugs, and future requests. We also substantially revised all the Jupyter Notebooks (not only to match the new data-transfer protocol) to help the final user to analyze the data obtained from the BrightEyes-TTM module.

• **Your vision on the availability of SPAD detectors – also other models – and how easy it is/would be to interface them to the proposed platform on one side and existing microscopes on the other.**

SPAD array detector will be the method of choice for fluorescence laser-scanning microscopy. This is our vision, but it is strongly supported by the microscopy community. Indeed, different European grants are supporting this vision and also the major microscopy companies are moving in this direction. This great interest led also to the foundation of two startups, namely Genoa Instruments from the IIT (Vicidomini G. holds shares of Genoa Instruments), and PI Imaging Technology from EPFL, whose missions are to sell and customize SPAD array detectors for laser-scanning microscopy – PI Imaging is also working on mega-pixel SPAD array detectors for wide-field microscopy. In short, in comparison to 2019 when the authors' lab demonstrated for the first time the benefits of the SPAD array detector for laser-scanning microscopy (Castello et al., Nat. Methods, 16:175–178, 2019), nowadays, these types of SPAD array detectors are readily available in the market.

Having an open-source time-tagging module does not only allows the use of current small SPAD array detectors, but it could become a standard to test the next generation of SPAD array detectors for laser-scanning microscopy. The possibility of developing ad-hoc FMC daughter cards to interface the module with the SPAD array detectors allows using the same module and analysis pipeline for different detectors. In the revised manuscript, we showed the compatibility of the module with both a 5x5 SPAD array prototype with TTL outputs and a commercial 7x7 SPAD array detector with LVDS outputs. In the future, we expect the blooming of new small SPAD array detectors with enhanced photon-detection efficiency (specifically in the near infrared-region), new geometries (e.g., linear SPAD arrays are fundamental for implementing spectral fluorescence lifetime microscopy), shorter dead-time, and higher temporal precision. Regarding the temporal precision, we think that current values are already optimal for all single-photon microscopy techniques developed so far, but we predict that many new single-photon microscopy techniques which may require higher temporal precision will be realized. As described above, with the current FPGA-technology and TDC architecture implemented, there is room to improve the performance of the module.

The module has been designed to be compatible not only with small SPAD array detector and custom microscopes, but also with existing commercial laser-scanning microscopes and conventional detectors. To stress the potential of the BrightEyes-TTM in the context of other detectors and (commercial) microscopy platforms, we performed several more measurements.

- Fig. S10: FLIM imaging with a commercial SPAD array detector (PRISM Light, Genoa Instruments, Italy) installed on a custom-built confocal setup.
- Fig. S13: FLIM imaging of Hela cells with α -tubulin immunolabelling and fluorescence lifetime measurements of a free dye with the BrightEyes-TTM on a Nikon confocal A1 system.
- Fig. S14: fluorescence lifetime measurement of a fluorescein-water solution with a commercial single-photon detector ($\text{\$PD-050-CTC-FC}$, Micro Photon Devices).

We changed the text in the manuscript accordingly to the above considerations. Specifically, we added the following main sentence(s):

“The principal aim of this work is to democratize SP-LSM by giving any microscopy laboratory the possibility to update its existing LSM systems. Together with a TTM, the other important ingredient to achieve this aim is the SPAD array detector. We demonstrate the compatibility of the BrightEyes-TTM not only with a SPAD array detector prototype but also with a commercial SPAD array detector. Currently, two startups are selling SPAD array detectors for SP-LSM (Genoa Instruments and Pi-Imaging Technology), but the interest from the major microscopy companies for this technology is growing: The AiryScan from Zeiss has become very popular and its transition to single photon-detector will be natural. On the other side, Genoa Instruments and Abberior Instruments already proposed integrating a SPAD array detector in their LSM product. Following this trend, we expect that also well-established detector companies will release a SPAD array detector designed explicitly for SP-LSM.”

- **Comparison with TDC state-of-the-art, e.g. in form of a table, ideally both research and commercial.**

As requested by the referee, we added two tables to the SI with a comparison between the BrightEyes-TTM and several other non-commercial Xilinx Kintex7 FPGA-based platforms (Supplementary Table S2) and commercial platforms (Supplementary Table S3). However, we would like to stress that:

- to the best of our knowledge, there does not exist an open-source project (hardware + software) with the aim to provide a time-tagging module to implement single-photon microscopy as well as conventional fluorescence lifetime microscopy and correlation spectroscopy;
- our aim was not to develop a new approach to improve the performance of state-of-the-art FPGA-based TDCs, but to implement a multi-channel TDCs optimal for the single-photon microscopy application — based on current and future SPAD array detectors for laser-scanning microscopy. Indeed, current state-of-the-art FPGA-based TDCs will represent a source of inspiration for the next version of the BrightEyes-TTM.

We changed the text in the manuscript accordingly to the above considerations. Specifically, we added the following main sentences:

In the main text. “In conclusion, the BrightEyes-TTM offers a combination of single-shot precision, linearity, temporal range, number of channels, and sustained count rate which is suitable for measuring fluorescence lifetimes with state-of-the-art SPAD array detectors for SP-LSM. It is worth noting that literature reports on different TDC implementations based on the same Kintex 7 FPGA family (Kintex 7) with superior characteristics (Supplementary Table S2). This aspect indicates the possibility of further improving the characteristics of our BrightEyes-TTM to match the expected reduction of jitter in the next SPAD array generations. In fact, it is important to observe that our current 25 channels TTM

implementation (as well as the 49 channels implementation of the next BrightEyes-TTM release) uses a small portion of the FPGA resources available (Supplementary Fig. S6), thus offering room for implementing strategies able to improve the overall characteristics of our TTM. For example, the single-shot precision of each channel can be improved by implementing a multichain measurement averaging method (21): The input signal of each channel is connected to multiple TDC simultaneously, and the final time is obtained by averaging the single TDC values.”

In the Discussions. “We implemented the TTM architecture on an FPGA, mounted on a readily available development kit, thus providing design flexibility and upgradability and also allowing fast prototyping and testing of new potential BrightEyes-TTM release versions. Compared to general-purpose commercial TTMs, our current BrightEyes-TTM version provides similar characteristics (linearity, temporal range, and sustained count rate) or partially inferior characteristics (temporal precision) but is still ideally suited for SP-LSM with current SPAD array detectors (Supplementary Table S3). Indeed, because our TTM has been specifically designed for fluorescence analysis, temporal constraints can be relaxed at the benefit of (i) compatibility with existing LSM systems, (ii) the possibility to integrate new features, and (iii) scalability in terms of photon channels. The BrightEyes-TTM supports 25 channels (i.e., the optimal number of elements requested for a SP detector array to implement ISM) in a single module. All commercial TTMs require multiple synchronized modules to support 25 channels. Furthermore, we anticipate the implementation of a new open-source BrightEyes-TTM version capable of handling up to 49 photon channels, which are useful for other SP-LSM techniques, such as S-FLIM.”

• **Calibration (p. 11): how often is a calibration necessary, how easy is it? Are there drift issues, e.g. with temperature changes, in particular for longer experiments, or as a function of the photon rates, and therefore TDC activity?**

We thank the reviewer for the comment. Every measurement is self-calibrated. However, to test the performance of the BrightEyes-TTM, we did an additional measurement where fluorescein was measured over a long-time span. Specifically, we performed a long measurement (more than 15 minutes) of a fluorescein-water solution (photon-flux 1 MHz) and plotted the phasor coordinates every minute. As shown in the figure below, the phasor coordinates are constant over time (the red square represents a zoom of the data points). During the measurement, the temperature of the FPGA was also monitored, and it resulted constant over time. Moreover, the stability of the BrightEyes-TTM was also tested in living cells (Fig. 6 in the main text), where we show the stability of the fluorescence lifetime over a longer acquisition.

- **Taking one step back: the Authors have made the basic assumption that SPAD arrays are the way to go, which I definitely share. However, they might need to better motivate their case.**

As mentioned in the introduction, SPAD detectors outperform conventional detectors for laser-scanning microscopy regarding time jitter, dynamics, robustness, and sensitivity. In the context of array detectors, SPAD arrays are scalable and less costly compared to other systems.

Besides a pure comparison between the characteristics of SPAD array detector with other well-established photo-detectors for laser-scanning microscopy (e.g. photomultiplier), we would like to stress the fact that SPAD array detectors are opening to a new microscopy paradigm able to provide new insights in many Life science open questions.

We think this claim is already well represented by the many publications (reported in the bibliography of the manuscript) of our groups and other groups (Gratton's Lab, Dan Oron's Lab, Lapkiewicz's Lab, Charbon's Lab, etc). Furthermore, (i) the major microscopy companies (Zeiss, Nikon, Picoquant, Abberior) installed or are planning to introduce a SPAD array detector in their confocal microscope; and (ii) new startups (Genoa Instruments and PI-Imaging Technology) started commercializing SPAD array detector for laser-scanning microscopy. To further stress this aspect, we added the following sentences in the Discussion:

“Together with a TTM, the other important ingredient to achieve this aim is the SPAD array detector. We demonstrate the compatibility of the BrightEyes-TTM not only with a SPAD array detector prototype but also with a commercial SPAD array detector. Currently, two startups are selling SPAD array detectors for SP-LSM (Genoa Instruments and Pi-Imaging Technology), but the interest from the major microscopy companies for this technology is growing: The AiryScan from Zeiss has become very popular and its transition to single photon-detector will be natural. On the other side, Genoa Instruments and Abberior Instruments already proposed integrated a SPAD array detector in their LSM product. Following this trend, we expect that also well-established detector companies will release a SPAD array detector specifically designed for SP-LSM.”

“We anticipate a future in which each single fluorescence photon will be tagged with a series of stamps describing not only its spatial and temporal properties, as shown in this work, but also its polarization and wavelength characteristics. A series of new algorithms -- based on conventional statistical analysis or machine learning techniques, and data formats -- will be developed to analyze and store such new multiparameter single-photon data-sets. As single-molecule techniques (38) have revolutionized cellular and molecular biology -- by observing molecule-by-molecule and not as an ensemble, we expect that recording the fluorescence signal photon-by-photon, while preserving most of the encoded information, can provide similar outcomes.”

DETAILED COMMENTS:

Main section (pp. 1-2): an introductory/overview image, e.g. featuring a general microscopy and detector set-up, could be really useful for the general reader, as well as a couple of lines explaining the SPAD fundamentals and related state-of-the-art.

The authors agree that an overview sketch showing a FLISM/FLFS setup and the data analysis protocol is useful and added the following figure to the main text, now Fig. 1.

Concepts of FLISM and FLFS. a A pulsed laser beam is focused and scanned across the sample. For each position of the laser beam, the fluorescence is collected and focused onto a SPAD array detector. Every photon produces a pulse in one of the detection channels almost instantaneously. The TTM measures the arrival time of the photon with respect to the last laser pulse and the pixel, line, and frame clock of the microscope. *b* A super-resolution fluorescence lifetime (FLISM) image can be reconstructed from the resulting 4D data set (x , y , t , ch). For each time-bin of the TCSPC histogram, a super-resolved ISM image is reconstructed with the adaptive pixel reassignment algorithm. All the images are then recombined and the fluorescence lifetimes are calculated for each pixel, resulting in the final FLISM image. *c* In spot-variation FLFS, the diffusion time as a function of the PSF width is measured. The PSF width can be changed by combining the photon traces coming from different detection channels. From the autocorrelations of the resulting intensity time-traces, the diffusion times, and hence the diffusion modality (free diffusion, diffusion through a meshwork, or diffusion in a sample comprising isolated microdomains), can be calculated. Simultaneously, from the start-stop times, the fluorescence lifetime is measured.

Results:

It might be useful to add, already at this level, a couple of sentences on the key TDC general aspects, and how their specifications were chosen as a function of the target SPAD array and/or application. (A description comes later, p. 8.)

The key general specifications of the BrightEyes-TTM project were chosen accordingly to the characteristics of the single-photon microscopy application, specifically on the characteristic of current SPAD array detectors used to implement single-photon microscopy. Since the timing precision of the currently available SPAD array detector (those used in this work and those available on the market) is in the range of 90-120 ps, we thought that a single-shot precision of 30 ps and a time-resolution of about 48 ps for our TDC was more than sufficient for carrying out time-resolved fluorescence experiments. Regarding the time-range of the TDC, as we wanted to have a flexible architecture able to perform any type of time-resolved measurement, we came up with the idea of using a monotonic counter to keep track of time. This approach yields no specific constraints when dealing with long time ranges. The only

limitation resides in the amount of data the hosting processing unit can store as the BrightEyes-TTM continuously streams time-sync data for being able to reconstruct the experiment time. Regarding the deadtime of our FPGA-based TDC, with a value of about 4.2 ns it is well below the current SPAD array detector dead-time (20-100 ns). A value of 4.2 ns is also smaller than the typical minimum laser repetition period commonly used in time-resolved fluorescence, i.e. 12.5 ns (80 MHz laser repetition rate).

We think this discussion fits better in the Methods section where the TDC architecture is fully described. However, to partially meet the request from the referee, we added the following sentences (in the Results section) in which we described the TDC characteristics in terms of the SPAD array characteristics:

“By fitting the start-stop time histogram with a Gaussian function, we estimated a precision (standard deviation of the fitted Gaussian distribution) of $\sigma = 30$ ps. We tuned the delay between the two signals across the whole temporal range of the fine TDC (here, 20 ns) and observed a similar precision for all the imposed delays (Supplementary Fig. S2), confirming the good linearity of the fine TDC. We repeated the same SSP experiment for the other channels and obtained a similar precision (Fig. S2 b). **This precision allows preserving the photon-timing precision of the SPAD array detectors used in this work (>90 ps both for the prototype and the commercial SPAD array detector).**”

“Lastly, we checked the sustained photon rate of the BrightEyes-TTM by implementing the SPP experiment for increasing photon rates (from 100 kHz to 50 MHz), keeping the SYNC signal at 50 MHz. The module starts saturating at approx. 15 MHz for a single channel and at approx. 5 MHz when all 25 channels received simultaneously a photon signal, which corresponds to a total photon flux of 125 MHz (Supplementary Fig. S3). In this work we used SPAD array detectors with 100 ns hold-off time, thus already limiting the sustained photon rate for each channel to 10 MHz.”

Maybe add one line highlighting the TDC results quality (e.g. excellent DNL/INL)

Together with the Table 2 and Table 3, where a fair comparison with many commercial and research time-tagging modules has been reported, we also added the following sentence:

In the main text. “In conclusion, the BrightEyes-TTM offers a combination of single-shot precision, linearity, temporal range, number of channels, and sustained count rate which is suitable for measuring fluorescence lifetime with state-of-the-art SPAD array detectors for SP-LSM. It is worth noting that literature reports on different TDC implementations based on the same Kintex 7 FPGA family (Kintex 7) with superior characteristics (Supplementary Table S2). This aspect indicates the possibility of further improving the characteristics of our BrightEyes-TTM to match the expected reduction of jitter in the next SPAD array generations. In fact, it is important to observe that our current 25 channels TTM implementation (as well as the 49 channels implementation of the next BrightEyes-TTM release) uses a small portion of the FPGA resources available (Supplementary Fig S6), thus offering room for implementing strategies able to improve the overall characteristics of our TTM. For example, the single-shot precision of each channel can be improved by implementing a multichain measurement averaging method (21): The input signal of each channel is connected to multiple TDC simultaneously, and the final time is obtained by averaging the single TDC values.”

Fig. 1f: if appropriate add a short comment on the points which fall outside of the universal semi-circle.

The amplitude of the phasor coordinates is affected by the signal-to-noise ratio (SNR). The lower the SNR is, the higher the offset in the histogram, and the lower the phasor amplitude will be. The authors explain this in more detail below in the answer regarding the comment on the uncorrelated background shifts.

Fig. 2: I would suggest to add a label on top of each column to make it easy to understand that the left one refers to “closed confocal”, etc., without the need to read the full caption, and possibly some other keywords if appropriate.

Please note that Fig. 2 is now Fig. 3. As suggested, we added the labels and keywords on top of each column. The image initially in panel (b) has been moved to the SI, Fig. S8. In addition, an image of primary mouse neurons was added to Fig. S8, showing the capability of the TTM to image samples that are more prone to photo-damage.

Fig. 2: which is the acquisition time? Is it possible to draw a comparison with what a “standard” AiryScan module would provide?

Single frame acquisition time for the Fig. 3 (Fig. 2 in the old version) is 26.2 s as the image has a dimension of 512x512 pixels and the pixel dwell time was set to 100 μ s. The authors added this information to the caption of the figure. The lowest dwell time for which the BrightEyes-TTM was tested was 10 μ s, which resulted in frame times of 2.62 s, 0.66 s and 0.164 s for image sizes of 512x512, 256x256, and 128x128 pixels, respectively. In our custom single-photon microscopes the major limitation for the scanning is represented by the galvo-mirror response time. We observed that 10 μ s is a reasonable value for not introducing motion artifacts in the recorded images. However, it is important to highlight that the only limitation introduced by the SPAD array detector is the dead-time of the single elements (100 ns), which precludes recording a second photon within this window. Indeed, in a previous work (Castello et al., Nat. Methods, 16:175–178, 2019), we use the SPAD array detector prototype in combination with a conventional photon counting DAQ and a resonant scanner system generating pixel dwell-time of few nanoseconds. Clearly, no more than one photon per pixel could be registered by this microscope implementation. However, the parallel nature of the SPAD array elements allowed for registering more than one photon per pixel. Because the BrightEyes-TTM can sustain 15 MHz per channel and 125 MHz in total, we believe the module is still compatible with this resonant-scanner implementation. On the other side, the AiryScan system speed is limited by the frame-like data transfer from the detector array, which is in the range of a few microseconds, thus fundamentally limiting the pixel-dwell time to this value and precluding the use with resonant-scanner microscopy architecture.

However, the fundamental limit for the speed of a laser-scanning system is the signal-to-noise ratio of the image. Within a nanosecond scale pixel dwell time, it would be challenging to collect enough photons to build up an image with a suitable SNR. For this reason, a very important parameter of the whole detection system is the so-called photon collection efficiency, i.e., how many photons reaching the detector are effectively recorded. This value depends on many characteristics of the detector (photon-collection efficiency, fill-factor, dead-time, etc.). While these characteristics are easy to find for the SPAD array detector used in this work, to the best of our knowledge, it is impossible to know for the AiryScan detector, making a fair and accurate comparison challenging to achieve.

The authors copied a table with the Zeiss Airyscan scanning performance below as a qualitative comparison.

Table Image Redacted

<https://www.zeiss.com/microscopy/int/products/confocal-microscopes/lsm-980.html#modes>

Is Fig. 2b clearly referenced in the text?

The authors made significant changes to the manuscript. As a result, Fig. 2b in the original version is now Fig. S8 (a). All figures are now clearly referenced in the text.

P. 4, uncorrelated background shifts: although it's not the main point, is it not possible to correct for this via background subtraction?

It is indeed possible to correct this shift by subtracting the background. The left panel of the figure below shows the raw phasor plot for a measurement of freely diffusing ATTO-495 for various laser powers. As expected, detector elements with a high SNR (such as the central element) show a higher amplitude compared to pixels closer to the border of the detector. Similarly, measurements with a higher laser power show a higher phasor amplitude.

For each channel, the background was calculated as the average value of the 50 lowest data points in the histogram. This value was subtracted from the histogram, negative values were set to zero. The background subtraction moves the phasor points closer to the universal circle, right panel. If the laser power is too low ($1 \mu\text{W}$), the histograms are dominated by background, and background filtering does not work well.

Figure 1: Example of the effect of the signal-to-background ratio on the phasor coordinates. (left) Phasor analysis on freely diffusing ATTO 495, concentration $22 \mu\text{M}$, for various laser powers, measured at the objective back focal plane. All 25 channels are plotted for each laser power. The highest power measurement was used as the reference measurement to account for the instrument response function of the microscope, detector, and TTM. The data points indicated with a black border are the central element

(square) and element 20, one of the corner elements (circle). The phase shifts between the measurements with different laser powers can be attributed to the shape of the laser pulse that differs slightly with the laser power. (right) Same data and plot as in (a) but with background subtraction. The background was calculated as the average value of the 50 lowest data points in the histogram. This value was subtracted from the histogram, negative values were set to zero.

Since the central message of the manuscript concerns the development of an open-source multichannel TTM and does not focus on how the data can be further processed in a second stage, the authors kept the original panel in the main text.

The authors changed the respective paragraph in the main text as follows:

“For this reason, the dark-noise (which appears as an uncorrelated background in the TCSPC histogram) increases with the quencher concentration. The same effect occurs on the phasor-plot: because the decays follow a single-exponential function caused by the collisional mechanism of the quenching, all points, regardless of the concentration, should lie on the universal semicircle. However, the uncorrelated background shifts the points towards the origin, since a lower signal-to-background ratio yields higher demodulation.”

o P. 5: where is Table 3?

The authors corrected the reference to the table, it is now Table S1 in the Supplementary Information.

• Methods:

o Fig. 4: the colours in the bottom left panel (light green and blue) are too similar.

The authors changed the color and line type in this panel (now panel (c) in Fig. 5) to make the curves easier to distinguish.

o P. 9: the total component cost is likely without the SPAD sensor itself, correct? Are there any additional costs due to licenses, e.g. software or firmware (such as proprietary IP blocks or libraries)?

All the license prices (Xilinx, Cypress) are included in the components cost. The price estimate for the BrightEyes-TTM hardware parts does not include the cost of the SPAD array detector.

o P. 10: custom PCBs: do they represent a potential obstacle for the open-source approach? Does every end-user need have them manufactured by him/herself?

Indeed, every end-user would need to manufacture the PCB him/herself. However, this should not be a major obstacle, as the authors provide in the online repository all the necessary information that the user would need to send to an electronic manufacturer that can assemble the PCB. For the end-user, this would not be much different from ordering any other component. As a group, we are right now offering the possibility to make a cumulative order for those people who already expressed some interest in the BrightEyes-TTM.

o Pp. 9-11: this long section is really a bit hard to follow without often resorting to the supporting images in the SuppMat -> can the situation be improved? Maybe by adding a “summary” image?

The authors added Fig. 1, which shows an overview of (some of) the methods that the BrightEyes-TTM can do. The authors rewrote the text accordingly and in a more accessible way.

o P. 12: “In this way, photons that are likely to not originate from the sample were removed.” -> can this be clarified?

Dark counts or background photons that do not come from the sample (e.g. room light) do not follow an exponential decay but instead contribute to an overall offset in the histograms. Within the filter boundaries, the filter values indicate how likely it is that a count is a photon originating from the sample. Outside of the filter boundaries, all counts can be considered dark counts/background and can thus be removed.

The authors changed the paragraph to better explain this:

“Having access to the start-stop time with ps range resolution, the BrightEyes-TTM allows classifying every photon in either of two classes: (i) almost certainly background or (ii) possibly fluorescence. In the former case, the photon can be completely removed from the data; in the latter case, the filter function is used to add a weight to the photon depending on how likely it is that the photon comes from the sample. Here, only counts in time bins between the peak of the histogram and about 10 ns later were included. The cropped TCSPC histogram of each channel was fitted with a single exponential decay function $H(t)=A\exp(-t/\tau_{\text{fl}})+B$, with amplitude A, lifetime τ_{fl} , and offset B a free fit parameters. The filters were calculated assuming a single exponential decay with amplitude 1 and lifetime τ_{fl} for the fluorescence histogram and a uniform distribution with value B/A for the background component. Then, for each count, a weight was assigned equal to the value of the fluorescence filter function at the corresponding start-stop time; counts that were detected directly after the laser pulse were assigned a higher weight than counts detected some time later, since the probability of a count being a fluorescence photon decreases with increasing start-stop time.

The second filter function describes the background component and was not used for further analysis.”

o P. 12: diffusion times and related results: do they match the state-of-the-art and/or your expectations?

Yes, they do. We already cited Slenders *et al.*, Biophys. Rep. **1**, (2021) and added Sadovsky *et al.*, Cell Rep. **18**, (2017) which report measurements of free-GFP in several cell lines. In addition, the results match the expected diffusion value given the dimension of eGFP.

The authors changed the paragraph slightly (p. 7, left column, red text):

“However, the average diffusion coefficient $D = 34 \pm 12 \mu\text{m}^2/\text{s}$, is well comparable with previous measurements of free eGFP in living cells (17, 29) and with the expected diffusion value knowing its dimension (MW = 27 kDa for eGFP).”

o P. 12: “afterpulse component”: how much is afterpulsing (and the other key SPAD parameters)?

The custom cooled SPAD array detector used for all measurements (except for Fig. S10, Fig. S13 and Fig. S14 where commercial detectors were used) has an afterpulsing probability of about 0.28%. Most pixels have a dark count rate of about 100 Hz and there is negligible cross-talk between any two pixels that are not nearest orthogonal neighbors. This detector is identical to the one used in Slenders *et al.*, Biophys. Rep. **1** (2), 2021, where more details can be found. The authors refer to Slenders *et al.* at the end of the paragraph *Laser Scanning Microscope* in the *Methods* section.

- **Bibliography:**

- o Some caps are missing, e.g. raman -> Raman, sted -> STED, etc

The authors made the necessary changes.

- **Bibliography – possible additional relevant literature:**

- o Recent review on ASIC TDC implementations for biophotonics applications: R. Scott, W. Jiang, and M. Jamal Deen, “CMOS Time-to-Digital Converters for Biomedical Imaging Applications”, IEEE Review Biomedical Engineering, 10.1109/RBME.2021.3092197

- o Q-ISM SPAD-based implementation: G. Lubin, R. Tenne, I. M. Antolovic, E. Charbon, C. Bruschini, and D. Oron, “Quantum correlation measurement with single photon avalanche diode arrays,” Optics Express, vol. 27, no. 23, Art. No. 23, Oct. 2019, doi: 10.1364/OE.27.032863.

- o Other FPGA-based multi-channel TDC implementation: S. Burri, C. Bruschini, and E. Charbon, “LinoSPAD: A Compact Linear SPAD Camera System with 64 FPGA-Based TDC Modules for Versatile 50 ps Resolution Time-Resolved Imaging,” Instruments, vol. 1, no. 1, Art. No. 1, 2017, doi: 10.3390/instruments1010006.

The authors thank the reviewer for the suggested literature. Lubin *et al.* was added to the introduction. In response to a comment from another reviewer, Scott *et al.* was added to the discussion. In addition, the authors added Ulku *et al.*, IEEE J. Sel. Top. Quant. (2019) to the discussion.

- **Supplementary:**

- o **P. 16: bin width: how small could you go in the case of (future generation) SPADs with much improved temporal resolution? Would you still be limited by design to a precision of ~30 ps?**

Assuming the same family 7 device from Xilinx and keeping all other hardware components identical to the ones used in this work, a temporal resolution (LSB) of 1.9 ps and a precision of 4.5 ps has been documented (Szplet et al, IEEE Transactions on Instrumentation and Measurement, 65(9):2088-2100, 2016) for the Kintex 7 FPGA. However, such extreme timing performances come at the cost of lower performances of other key parameters, such as the dead time (87.7 ns) and the maximum number of deployed channels (8). In future generations, depending on the SPAD detector characteristics and the application, one can choose to improve the precision, keeping in mind this trade-off.

- o **P. 17: which is the relationship between the histogram bin width of 48+-3 ps, and the previously reported 48+-25 ps?**

Assuming the referee means 48+-3 ps and 43+-25, the value 48+- 3 ps represents the average time-width of the start-stop time histogram obtained from the code density test experiment and the DNL analysis for a specific channel (here, channel #12). Indeed, when performing a statistical code density test, the reconstructed start-stop time histogram should ideally be a constant flat line, indicating that every time-bin has an equal time-width within the measured temporal range. Specifically, we measured a sigma-value equal to 0.06 LSB from the start-stop time histogram. Since, from the calibration, we imposed an LSB equal to 48 ps, the sigma-value in ps is equal to 2.88 ps, thus the average time-width is equal to 48+-3 ps.

The value 43 ± 25 ps is similarly obtained from a code density test experiment but from the bin-to-bin calibration analysis. It represents the average time-width of the CARRY4 used within the tapped delay lines (i.e., TDC). Furthermore, it represents the average across the whole channels of the BrightEyes-TTM.

Since in the new version we have 25 channels (not anymore 21), we repeated the experiment, and we obtained a value of 43 ± 16 ps (see Supplementary Fig. S29).

o P. 18: Fig. S1b: is the linearity unit (i.e. arbitrary units) correct?

The authors thank the reviewer for noticing this mistake. Indeed, the y axis of this panel, now Fig. S2 (b) is ns.

o P. 39: Fig. S22: how are the outliers taken into account?

Fig. S22 is now Fig. S29. The outliers were included in the calculation of the average time-width value. That is why the calculation shows a standard deviation of ± 16 ps.

TYPOS (non-exhaustive list):

(Fig. 1) pane -> panel?, (p. 7) "version able to handle up to 49 photon channels Whilst" -> full stop missing, "course counter" -> coarse, (p. 8 right 4th line from the bottom) "t_sysclk" or "T_sysclk"?, (p. 9) "temporally correlated" or uncorrelated?, "tap-delay line" -> tapped-delay line? (or tapped delay line?) , "the photons-signals", "Since the TTM architecture uses a start-stop reverse strategy to reconstruct the photon start-stop time – for a given channel ch, it is..." -> unconventional writing, maybe rephrase, CARR4 -> CARRY4, "The inversion of these values" -> reciprocal?, "recording the simultaneously the signal", "fingerprint ans summed" -> and.

The authors thank the reviewer for the feedback. The authors carefully read the manuscript again and made the necessary corrections. Since many changes were made, only the main changes are indicated in red.

Reviewer #2 (Remarks to the Author):

Rosetta et al. report on the development of an open-sources multi-channel time-tagging module designed to implement and democratise current and future fluorescence single photon laser scanning microscopy (SP-LSM) techniques. They claim that their FPGA-based approach enables quick prototyping, updating and adaptation. The module was integrated in an existing confocal laser scanning microscope equipped with a 5x5 SPAD array to perform fluorescence lifetime image scanning microscopy (FLISM). The authors also claim that for the first time they demonstrate the combination of comprehensive correlation analysis (CCA) with fluorescence lifetime analysis using the diffusion mode of eGFP with its lifetime. They provide detailed guidelines, hardware parts lists and open-source code for FPGA firmware and operational software along with their manuscript.

Overall, I find this manuscript quite interesting and possibly suited for publication in Nat. Commun. after some major revisions as detailed below.

The authors thank the reviewer for the positive evaluation of the manuscript. The authors made several major and minor changes; a detailed description of all changes is written below in the point-to-point responses to the reviewer's questions. The authors believe that having incorporated the concerns raised by the referee, the quality of the manuscript has substantially improved.

Major comments

1) p4, right column, 1st paragraph: The authors explain the deviation from the expected universal semicircle in the phasor plot by uncorrelated background. While this might be the most suitable explanation, it is not clear if it is the only explanation. Could this be proven e.g., by comparison with a different dye?

The authors performed lifetime measurements on freely diffusing ATTO495 and compared different SNRs by changing the laser power and comparing different detector elements of the SPAD array.

Figure 1: Example of the effect of the signal-to-background ratio on the phasor coordinates. (left) Phasor analysis on freely diffusing ATTO 495, concentration 22 μM , for various laser powers, measured at the objective back focal plane. All 25 channels are plotted for each laser power. The highest power measurement was used as the reference measurement to account for the instrument response function of the microscope, detector, and TTM. The data points indicated with a black border are the central element (square) and element 20, one of the corner elements (circle). The phase shifts between the measurements with different laser powers can be attributed to the shape of the laser pulse that differs slightly with the laser power. (right) Same data and plot as in (a) but with background subtraction. The background was

calculated as the average value of the 50 lowest data points in the histogram. This value was subtracted from the histogram, negative values were set to zero.

2) On p5, the authors discuss state the use of the phasor-based segmentation. What motivates the use of phasor-based segmentation? Can the authors please discuss possible advantages/disadvantages over the lifetime threshold-based approaches?

Phasor based segmentation is fast (since no nonlinear fitting is needed) and model-free (so no assumption on the number of species has to be made), simplifying the analysis of large datasets and making the method easily accessible to non-experts. The coordinates in the phasor space give information on the lifetime and the number of components. By checking for clusters of pixels in the phasor space, one can derive if the image contains two (or more) molecular species and map these in the image space.

The BrightEyes-TTM gives access to the start-stop time of every photon. The user can then choose which type of analysis to do in post-processing: lifetime fitting and phasor analysis are two examples shown by the authors in several figures. The segmentation in Fig. 4 was done with the phasor approach to demonstrate how different phasor coordinates are back-projected to different structures of the cell.

3) The authors mention different measurements using the B&H TCSPC system. However, from the shown data, the reader can't decide what they will gain/lose when switching from one system to the other.

The B&H DPC-230 is based on ASIC technology implementing a TAC device, meaning that it cannot be further modified nor upgraded like the FPGA technology. Furthermore, the B&H DPC-230 has a worse time resolution and jitter precision compared to the BrightEyes-TTM. Lastly, due to the limited number of input channels, the DPC-230 cannot be used in conjunction with a 25 element SPAD array detector.

We used the B&H DPC-230 to roughly validate our time-tagging module with respect to a commercial system that was the standard in our lab and in many other labs. More in general, the main scope of this work is not providing a system that is superior in performance with respect to all current time-tagging modules commercially available but triggering an open-source project on this topic. No doubt that single-photon microscopy in general and fluorescence-lifetime imaging, in particular, are living a great momentum now.

4) Can the authors please provide a clear concise benchmark comparing BrightEyes with, for example, the B&H TCSPC system?

The authors added the following information in the Table S3.

Parameter	BE-TTM	B&H DPC230 ¹
No. of channels	25+1 tested (49+1 test ongoing)	16
Input type	LVC MOS 25 and LVDS	LV TTL
Min. Input pulse width	4.2 ns	2 ns
Time bin width	user defined (default 43 ps)	164.61 ps
Single shot precision	30 ps	Not specified
Dead time	4.2 ns	< 10 ns
Imaging	Scanning, recording of Pixel, Line, and Frame Clock Pulses	Scanning, recording of Pixel, Line, and Frame Clock Pulses
PC interface	USB 3.0 SuperSpeed	PCI

Maximum laser sync rate	80 MHz	150 MHz
Differential non-linearity	~ 6 % RMS	Not specified
Inputs connector type	SMA	SMA, MCX
Time range	not limited by hardware - Tested at 200 ns (5 MHz), 100 ns (10 MHz), 50 ns (20 MHz), 25 ns (40 MHz), 12.5 ns (80 MHz)	Not specified
Max. Pixel Rate	100 KHz	2 MHz
Readout	continuous readout during measurement	continuous readout during measurement
Data Acquisition mode	Single-photon time tag and its associated reversed start-stop time in respect with the laser sync	Absolute Time Mode TCSPC Mode Multiscaler Mode
PC System requirements	Linux (native) / Windows (ported) min. 1.5 GHz CPU clock, min. 16 GB RAM memory, SSD hard disk	Windows 8 / 10, > 8 GB RAM, 64 bit operating system recommended
Sustained Readout Rate	7.5 * 10 ⁶ photons	7 * 10 ⁶ photons
Dimensions	Approx 240mm x 270mm x 40 mm	312 mm x 130 mm x 15 mm
Price	3000 €	15000 €

¹ <https://www.becker-hickl.com/products/dpc-230/>

A big limitation of the B&H DPC230 in the context of single-photon microscopy is that when using all 16 channels it is not possible to filter the sync signals that are not followed by a photon signal, making it not realistic to use with an 80 MHz pulsed laser system.

5) Furthermore, I would be interested in the experimental saturation limit of the systems? Is it completely independent from label density/brightness?

This is a very important point that we missed in the first version of the manuscript. The authors thank the reviewer for pointing it out. The authors added a new figure, Fig. S3, which shows the experimentally observed count rate as a function of the imposed count rate. The Bright-Eyes TTM can sustain an overall count rate of about 125 MHz when all 25 channels are active, with a maximum count rate per channel of about 5 MHz. If only one channel is used, the maximum count rate may be up to around 15 MHz. Comparing this value with a commercial system, such as the Becker and Hickl DPC230 card, Table S2, one can conclude that the BrightEyes-TTM performs very similar in this respect. The Becker and Hickl DPC230 card implements an event filter (i.e., the sync signal is registered only after a photon signal is detected) only when 4 channels are measured. Again, our scope is not to compare our module with the Becker and Hickl DPC230. For this reason, we did not include this discussion in the main-text, but we use it only to reply to the referee.

Fig. S3. Sustained count rate.

6) The data shown in fig 2 compares in vitro data with data in fixed cells. It is not clear from the text if the differences observed for the lifetime histograms are due to different fluorescent labels or different chemical environments affecting the same fluorophore. Furthermore it is not clear if there is a quality difference between the data from the phantom and the cell experiments. How does sample preparation affect the results? Is there anything interested labs will have to account for when choosing the BrightEyes approach?

To demonstrate that the BrightEyes TTM can also be used in more fragile samples such as living neurons, the authors added a panel, panel (b) in Fig. S8, with an image of primary mouse neurons expressing (SEP)-tagged-β3 subunit of the GABAA receptors. No special sample preparation is needed compared to conventional confocal microscopy or FLIM.

7) Fig 5f shows different measurements (grey lines). However, the associated mean seems to have a significant bias (shifted downwards in resp. of the lifetime axis). Can the authors please explain/consolidate this?

The author thanks the reviewer for the comment. In this subplot, there has been a mistake while preparing the figure. The authors made the proper changes and carefully checked the entire manuscript.

8) In the discussion of the results shown in fig 5, readers might find it difficult to understand the relationship between mobility and the diffusion coefficients at different scales. I strongly recommend a clearer explanation for broader audience in Nat. Commun.

The authors added a new figure, now Fig. 1, which contains introductory sketches on the setup and the different types of analysis mentioned in the manuscript. Many changes were made in the text to adapt the writing style to the broader audience of Nat. Comm.

9) As the authors claim easy adaptation for future systems, they should include the limitations of their approach when it comes to larger SPAD arrays. Will this architecture allow to deal with SPAD cameras? What will be required to do this and could their approach be extended in that direction?

The BrightEyes-TTM can indeed be adapted for future systems. Although in the current manuscript, a 5x5 SPAD array detector was used for most measurements, the authors anticipate the implementation of a new version capable of handling up to 49 photon channels. Notably, the asynchronous read-out scheme of individual photons becomes more complex for a higher number of channels. The BrightEyes-TTM is designed for platforms such as laser-scanning microscopes where the number of detector elements is limited. In particular, the amount of data that has to be transferred (and stored and analyzed) will become the main bottleneck for bigger detectors. Clearly, the platform cannot be used for large SPAD cameras with thousands of pixels.

The authors added more explicitly the limitations of the BrightEyes-TTM to the discussion, p. 8, left column:

“Whilst the time-tagging module is the most informative DAQ system, it requires transferring (to the PC) and storing a consistent amount of data. Clearly, this approach is not scalable to large SPAD cameras. Depending on the application, this problem can be reduced by performing part of the data analysis and the reconstruction directly on the TTM, by leveraging a novel class of DAQ cards equipped with ARM-based processors. Alternatively, an application-specific integrated circuit (ASIC) TDC implementation can be used for large SPAD imagers with thousands of pixels.”

10) The authors claim that their system enables quick prototyping, updating and adaption. However, on p 10 they mention a LabView control system that seem not be documented in their data.

The LabVIEW control system was not specifically designed for the current project but was already used before, see Slenders *et al.*, *Biophys. Rep.* **1**, (2021), and based on the Carma software that is described in the references mentioned in the paragraph *Control system* in the *Laser Scanning Microscope* section of the methods. The LabVIEW control system sends out pixel, line, and frame clocks and drives the scan mirrors while recording the signal from one or more elements of the SPAD array detector. The BrightEyes-TTM works as a slave, it only needs input from the detector and the timings from the different clocks (pixels, lines, frames). To demonstrate this, the authors also used the TTM with a commercial Nikon microscope, Fig. S14.

As advocates of open science, the authors are working on also making the control software, called BrightEyes-CS, open-source.

11) On p10 the authors describe a fail-safe mode. However, it remains unclear to me if/how this is reflected in the data, so that the user can decide on how to proceed with it.

When the photon flux is too high, the TTM temporarily enters a fail-safe mode, in which priority is given to the pixel, line, and frame flags and to the force-write events. This strategy ensures that all photons that are collected when the TTM switches back to the normal mode can be registered correctly, i.e., with the absolute arrival times, and that only the photons that arrived when the fail-safe mode was active are lost. After the measurement, the user can check if the measurement has entered the fail-safe mode. If this is the case, the user can choose to proceed with the data, knowing that the data is incomplete, or e.g., decide to repeat the measurement by reducing the photon flux, for example, by reducing the laser power. However, we would like to stress that the saturation limit of the module is compatible with most of the experiments performed in our lab (see the above discussion on the module saturation level), which makes this event very rare.

Minor comments

12) Can the authors explain the weak tendency for longer lifetimes in fig. 1e as compared to B&H (before B&H signal has cut-off)?

As the B&H DPC-230 is based on TAC technology, its response becomes nonlinear at the beginning and at the very end of the probed time range. This non-linearity affects the overall TCSPC histogram reconstruction. That is why the B&H response differs from the BrightEyes-TTM for longer lifetimes, as it already shows nonlinear behavior even before the cut-off region. Cut-off regions and proximal non-linear behaviors are embedded and intrinsic features for TAC-based time-tagging devices.

13) Are the arrows in fig 1d supposed to indicate FWHM (don't hope so)?

The authors removed the arrows in the figure, now Fig. 2 (c), to avoid confusion. The FWHM of the systems are 250 ps for the B&H system and 200 ps for the BrightEyes-TTM. The authors added the values in the caption of the figure.

14) Can the authors explain the clear deviation of the INL from an equal distribution in fig 1a?

The INL profile is very irregular and deviates from an equal distribution as the different CARRY4 elements that compose the tapped-delay-line structure have different delay-values. The delay value discrepancies ultimately depend on small anomalies in the silicon fabrication process. The INL depends on the intrinsic FPGA fabric characteristics and may differ from card to card. Another KC705 card may show different behavior. That is mainly why the bin-by-bin timing calibration is strongly needed when pursuing a tapped-delay-line approach. The authors have no explanation for this specific behavior of the used card

The INL depends on the intrinsic FPGA fabric characteristics and may differ from card to card. Another KC705 card may show a different behavior. The authors have no explanation for this specific behavior of the used card.

15) p4, left column, first paragraph: Can the authors please indicate in fig 1d the “additional bump” they discuss in the explanation?

The additional bump can be attributed to the effect of optical crosstalk between adjacent pixels of the SPAD array detector, as shown in the following publication record:

See Fig. 6 in “Mauro Buttafava, Federica Villa, Marco Castello, Giorgio Tortarolo, Enrico Conca, Mirko Sanzaro, Simonluca Piazza, Paolo Bianchini, Alberto Diaspro, Franco Zappa, Giuseppe Vicidomini, and Alberto Tosi, *SPAD-based asynchronous-readout array detectors for image-scanning microscopy*, *Optica* **7**, 755-765 (2020).”

The authors refer to this publication in the corresponding paragraph. For clarity, the figure below shows the bump in the inset (zoomed in).

16) In the discussion of fig 4c (p5) the authors state a zero intercept of the lifetime dependency. However, fig 4c shows an intercept close to zero. Can the authors please account for this deviation?

The intercept with the y axis is zero for free diffusion. The curve shown in the figure, now Fig. 5 (e) is a linear fit ($y = ax + b$) of experimental data with both a and b as free parameters. As a result, the curve itself does not exactly pass through the origin, but taken into account the uncertainty of the measurement, one can conclude that the diffusion is free. E.g. for the unfiltered case, from the covariance matrix returned by the fit, a standard error of 0.09 ms is found for b . Hence, $b = (0.09 \pm 0.09)$ ms.

17) Fig S1d, am I mistaken or does the calculate standard deviation show weak correlations? Can the authors please look into this and explain?

The standard deviation of the single shot precision plots (Fig S1d) but also (Fig S1c, Fig S1b) shows a temporal correlation with a period of 2.5 ns (equivalent to a frequency of 400 MHz). This frequency is not a clock from BrightEyes-TTM, but it corresponds to the central clock of the signal generator (SYLAP). To generate the signals, it uses the main clock at 400 MHz and FPGA elements provided from Xilinx called "IDELAY," which allow generating signals with delays with a granularity of $1/64^{\text{th}}$ of main clock frequency (400 MHz), which is 39.0625 ps.

18) Can the authors please briefly explain why in fig S3, the resolution in cells is smaller than in the phantom sample?

The resolution measured by FRC analysis is not only sensitive to the optical setup but also depends on other aspects such as the SNR. As such, different values will be found for different samples, pixel dwell times, laser powers, etc. It is not uncommon to measure a better resolution in a bright and flat phantom beads sample than in a more complex biological sample, see, e.g. Slenders *et al.*, Biophys. Rep. **1**, (2021).

Note that many major changes were made to the manuscript; the original Fig. 2 (now Fig. 3) only shows the beads sample. Consequently, the FRC analysis of the cell sample was removed from the SI.

potential typos:

19) Fig 1 – figure caption, line 3: “differential non--lineraity”

p2, right column, last paragraph: “(the 4 corner element are neglected)”

20) p7, right column, 1st paragraph: “... handle up to 49 photon channels Whilst, time-tagging module ...”

21) p10, right column, 2nd paragraph: the last sentence seems to be incorrect.

The authors carefully read the manuscript again and made the necessary corrections. Since many changes were made, only the main changes are indicated in red.

Reviewer #3 (Remarks to the Author):

Rosetta et al. present a technical tour-de-force on a homebuilt multichannel time tagging module called BrightEyes-TTM. They provide extensive characterization and a few application examples using a SPAD array. The methods and supplemental information are thorough. I would feel comfortable building a BrightEyes-TTM unit if we owned a SPAD array.

Before I can provide a final recommendation, I suggest that the authors streamline the manuscript to give the reader a clearer picture of the accomplishments. For example, is the focus the technical innovation, the experiments that BrightEyes-TTM enables, or both? Further controls and modeling may be required if the focus is to present new experiments. There also needs to be a more straightforward presentation of the technical points in both the writing and figures.

Here I provide a set of major and minor questions/notes and suggest potential actions for each.

The authors thank the reviewer for appreciating this technically challenging work and for the feedback. The authors also appreciate the fact that the referee signed the review.

The authors performed several extra measurements and made substantial changes along the whole manuscript to address the referee's comments, suggestions, and doubts. A point-by-point answers can be found below. However, before the point-to-point responses, we would like to give the referee a general overview of how we approached the revision.

Since the beginning, the main scope of this work was to provide to the "microscope makers" (or DIY microscopy community) an open-source data-acquisition module to implement what we call single-photon microscopy. In short, a laser-scanning microscope is able to register the fluorescent signal photon-by-photon and with a series of tags able to encode extra sample's information. Our group and other groups use this information for implementing novel imaging and spectroscopy techniques such as ISM, FLISM, Q-ISM, SOFISM and CCA. We believe that this is just the tip of the iceberg – we are sure that the referee agrees with that – and that many more new methods will be developed, hence the decision for an open-source project.

Successively, as requested by the Editor in response to the manuscript pre-inquiry, we extensively validated the BrightEyes-TTM, not only by implementing a well-established single-photon microscopy technique, such as fluorescence lifetime image scanning microscopy (FLISM), but also by setting the base for a brand-new technique -- named fluorescence lifetime fluctuation spectroscopy (FLFS), which combines the benefits of CCA and fluorescence lifetime analysis to decipher bio-molecular processes inside living cell. Although we strongly believe in the great future of FLFS in the original manuscript, and in this revision, we decided to keep the focus on the technical innovation provided by the BrightEyes-TTM. Indeed, we are working on a new methodological work that focuses only on FLFS, showing the benefits of its application to study RNA-binding protein dynamics and synaptic protein dynamics. In short, we think that a deep analysis of FLFS is outside of the scope of this paper. On the other side, as suggested by all referees we implemented a series of new experiments and changes in the BrightEyes-TTM to reinforce the original and main goal of the manuscript, i.e., democratizing single-photon microscopy. Furthermore, thanks to the referees' comments, we realize that this mission could be achieved only starting from the democratization of conventional FLIM.

In this context, for the revision we:

- showed the compatibility of our BrightEyes-TTM with a commercial 7x7 SPAD array detector (PRISM Light detector LVDS outputs, Genoa Instruments). This is also a demonstration of the versatility of the module. Indeed, the commercial SPAD array detector uses an LVDS output standard, which is different from the TTL outputs of the 5x5 SPAD array prototype used so far. Thus, we develop a new custom I/Os daughter card connected via the FPGA mezzanine connector (FMC), whose grab file is shared on the BrightEyes-TTM new ReadTheDocs pages. This demonstrates how in the future the module can also be used for new SPAD array detectors, which indeed will be developed to further improve their characteristics.
- redesigned the data-transfer (to the computer) protocol and system to make the scalability of the module much easier. In the reviewed manuscript, we present the new 25-channel version (the old version provides only up to 21-channels), and in the ReadTheDocs pages, it is already possible to find the 49-channel version, which is under test.
- validated the module with conventional single-photon detectors, a commercial single-element SPAD detector, and a photomultiplier whose signal has been digitalized with a constant fraction discriminator. The authors are aware that to effectively create a large community of users for the BrightEyes-TTM and for the corresponding analysis software, also the laboratories in which a SPAD array detector is not available, can readily implement conventional FLIM or fluorescence-lifetime-correlation spectroscopy.
- used the module to transform a commercial laser-scanning microscope into a FLIM microscope. This demonstrates the versatility of the module to work also with commercial microscope control-units. The only requirement is that the control-unit provides as outputs the scanning synchronization signals.
- created a ReadTheDocs web page linked to the GitHub repository to host the documentation for the open-source project.

Major

1. Figure presentation.

All of the figures are complicated to follow. For example, Figure 4 contains a large amount of information presented on two different measurements, fluorescent beads, and Alexa488. In addition, the data is overwhelming, making it difficult to understand the results and how the BrightEyes-TTM architecture enabled the work.

Here I provide some suggestions for each figure, intending to improve readability:

Figure 1. The cartoons in the top row are helpful to guide the reader. It would be helpful to provide a clear explanation of why each test is essential in the text. For example, the explanation on testing linearity is highly technical (page 3, first paragraph of left column). Why is this test important in the context of biological imaging? The same goes for all of the tests shown in Figure 1.

For a general audience journal, I suggest that some of the finer details of Figure 1 can be moved to the supplement and replaced with cartoon explanations in the main manuscript.

The authors follow the comment of the reviewer and decided to make the main text and main figures less technical to make the paper more interesting for a general audience. As such, the authors simplified the figure, now Fig. 2, keeping the most essential aspects in this figure, and moving the other panels to the SI,

Fig. S1 and Fig. S2. Fig. 1 in the new version of the manuscript contains cartoon explanations showing the concepts of FLISM and FLFS with the BrightEyes-TTM. The text was changed accordingly, with the main changes colored in red.

Figure 2. The difference between the columns is not clear from looking at the figure. However, a simple cartoon showing the configuration would emphasize why the far left column has low signal-to-noise but similar shapes to the column to the right (0.2 AU vs. ISM).

The authors added column titles to make the figure, now Fig. 3, better readable. Panel (b) has been moved to the SI, now Fig. S8 (a), together with a new measurement on primary mouse neurons, Fig. S8 (b).

The resolution increase is not apparent due to the lookup tables and saturation choices. Please present averaged bead profiles, with average FWHM and average FRC results on the vimentin in addition to cleaner intensity images.

The author added a line profile of a fluorescent bead in the Supplementary Fig. S7 for the confocal and the APR-ISM images (on the right). In the case of the ISM image there is a significant improvement of the signal-to-noise ratio in comparison to the confocal case (about a factor 7). While, the resolution does not significantly improve between the two cases by calculating the FWHM (239 ± 11 nm for confocal and for 245 ± 5 nm for APR-ISM, N = 7), the precision of the resolution improves in the ISM case, as the signal-to-noise ratio is higher.

Please explain how the rejection criteria for the lifetime histograms are determined. They appear aggressive for the vimentin. Why is this necessary?

For the vimentin image, the lifetime histogram bounds are symmetrical around the peak in the FLISM histogram (central plot). The width was chosen by visually inspecting the FLISM image and maximizing the contrast. The same bounds were then used for the confocal (0.2 AU) and open confocal (1.4 AU) data. The histogram bounds for the other samples were also chosen by visually inspecting the FLISM image and applying the same bounds to the confocal images.

The authors added the following sentences to caption of the main figure, now Fig. 3:

“The selected interval was chosen by visually inspecting the FLISM image. The same intervals were used for the confocal and open confocal data.”

The authors added the following sentences to caption of Fig. S8:

“For (a), the lifetime histogram bounds are symmetrical around the peak in the FLISM histogram. The width was chosen by visually inspecting the FLISM image and maximizing the contrast. The same bounds were used for the confocal and open confocal data. For (b), the histogram bounds were chosen by visually inspecting the FLISM image and applying the same bounds to the confocal images.”

Why are the lifetime distributions significantly different between the 0.2 AU and ISM/1.4 AU data? For the beads, this appears to be due to fewer photons. However, the distribution appears significantly broader for the vimentin with a shifted peak. Would the lifetime histograms match if the experiments collected the same number of photons?

The lifetime distributions are strongly influenced by the signal-to-noise ratio. In the case of confocal imaging, the number of photons is low, affecting both the peak position and the peak width. The authors

added another example with primary mouse neurons, Fig. S8 (b), giving a similar conclusion. The vimentin figure was moved from the main text to Fig. S8 (a).

How is Δt handled for the ISM data? It is not completely clear to me from the methods and supplemental material. Specifically, during photon reassignment, are there technical issues that must be accounted for to ensure the arrival time is correct?

The authors added a new figure, now Fig. 1, in which the concepts of FLISM and FLFS are described. As shown in panel (b), an ISM image is made for each time bin. In a second step, all ISM images are combined to make a FLISM image. A more detailed explanation of the algorithm is reported in Supplementary Fig. S30. Notably, the same algorithm has also been used in the original FLISM paper (Castello et al., Nat. Methods, 16:175–178, 2019).

Because each photon channel is implanted with a different TDC (i.e., a different tapped delay-line), each photon channel has a deterministic delay, which must be compensated before each analysis. To align the histograms of the different channels, a reference measurement with freely diffusing fluorescein was used. The authors added the following to the *Application-dependent analysis* paragraph in the *Data analysis* section:

“Importantly, because each SPAD element and each TDC can introduce a different, but fixed, delay, it is necessary to temporally align the histograms of the different channels: We performed a reference measurement under identical conditions with freely diffusing fluorescein. The resulting histograms were plotted in the phasor space. The phase difference between the expected position in the phasor space (given the known lifetime of 4.1 ns of fluorescein) and the measured position for each channel was used to calibrate each channel, i.e., to shift the histograms of the actual measurement back to the correct position.”

Note that this temporal-alignment procedure is typical for any multi-channel time-tagging module and analysis type (i.e., FLISM, FLCS, FLIM). Our BrightEyes-TTM is not immune. Furthermore, it is necessary for any analysis.

Fig. 1. Concepts of FLISM and FLFS. **a** A pulsed laser beam is focused and scanned across the sample. For each position of the laser beam, the fluorescence is collected and focused onto a SPAD array detector.

*Every photon produces a pulse in one of the detection channels almost instantaneously. The TTM measures the arrival time of the photon with respect to the last laser pulse and the pixel, line, and frame clock of the microscope. **b** A super-resolution fluorescence lifetime (FLISM) image can be reconstructed from the resulting 4D data set (x, y, t, ch). For each time-bin of the TCSPC histogram, a super-resolved ISM image is reconstructed with the adaptive pixel reassignment algorithm. All the images are then recombined and the fluorescence lifetimes are calculated for each pixel, resulting in the final FLISM image. **c** In spot-variation FLFS, the diffusion time as a function of the PSF width is measured. The PSF width can be changed by combining the photon traces coming from different detection channels. From the autocorrelations of the resulting intensity time-traces, the diffusion times, and hence the diffusion modality (free diffusion, diffusion through a meshwork, or diffusion in a sample comprising isolated microdomains), can be calculated. Simultaneously, from the start-stop times, the fluorescence lifetime is measured.*

Figure 3. This is a great experiment. Unfortunately, the ISM images appear over-saturated, rendering the ISM contrast and resolution less impressive than it is. Please correct.

The authors updated the figure, now Fig. 4.

How is the segmentation performed on the phasor plot? Please describe the procedure in more detail. For example, how is the center and radius of the circles for each dye state determined? From another experiment or theory?

The two components are separated by a line that passes through the origin and the center of mass of the phasor coordinates. Since the lifetime is encoded in the phase of the phasor coordinates, the points below this line correspond to pixels with a relatively short lifetime, the points above the line with a longer lifetime. The authors changed the phasor plot in Fig. 4 to better describe the segmentation between long and short fluorescence lifetimes.

Phasor plot representation and analysis for lifetime are becoming more and more popular, and we are aware that more robust methods can be implemented to unmix molecular components based on their lifetime (e.g., Alex Vallmitjana, Belén Torrado, and Enrico Gratton, "Phasor-based image segmentation: machine learning clustering techniques," *Biomed. Opt. Express* 12, 3410-3422 (2021)). However, in our experiment we did not expect a clear lifetime separation as for a multi-probes experiment (i.e., each probe has a different fluorescence lifetime), but rather a continuous distribution. For this reason, and since also the aim of the experiment is to validate the ability of the BrightEyes-TTM to reveal lifetime changes, we implemented a very simple segmentation method.

Because this figure focuses on live-cell data, it would be helpful to show dynamic information. For example, a supplemental movie showing the membrane dynamics over time using ISM and FLISM side by side would be a powerful visualization.

The authors added a movie in the supplementary information (Fig. S31 shows the first time point) depicting simultaneously the FLISM and the ISM images of HeLa cells stained with the membrane sensitive probe ANEP.

Figure 4. Figure 4 has an overwhelming amount of information. A cartoon of circular scanning FLFS, why this removes the need for calibration, and why high background can influence the correlation function will help guide the reader.

The authors moved the plots on the diffusing goat anti-mouse antibodies to the SI, simplified the other panels, and added a sketch on the principle of scanning-FCS. The new figure with caption is shown below.

Fig. 5: BrightEyes-TTM for circular-scanning FLFS on freely diffusing fluorescent beads. (a) Sketch of the concept of circular-scanning FLFS. A pulsed laser beam is scanned in circles of radius R (top panel) while both the absolute photon arrival times (center-left panel) and the start-stop times (center-right panel) are registered. The autocorrelation function of the intensity trace is calculated, from which simultaneously the size of the focal spot w_0 and the diffusion time t_D can be extracted. (b) Start-stop time histograms for the different pixels, bin width 48 ps, total measurement time 226 s, central pixel in black. (c) Exemplary filter functions for the central pixel data. (d) Autocorrelations and fits for the central pixel, sum 3x3, and sum 5x5, for the unfiltered (left) and filtered (right) case. (e) Diffusion time as a function of w_0^2 (left) and average number of particles in the focal volume as a function of the focal volume (right). The corresponding diffusion coefficients are $(14.3 \pm 0.5) \mu\text{m}^2/\text{s}$ (unfiltered) and $(14.0 \pm 0.4) \mu\text{m}^2/\text{s}$ (filtered). The fitted particle concentrations are $(7 \pm 3) / \mu\text{m}^3$ (unfiltered) and $(1.70 \pm 0.03) / \mu\text{m}^3$ (filtered).

The rationale behind the filtering used to account for detector afterpulsing and background signal is unclear. Can the authors please expand on this in the methods and connect it to their benchmarked performance in Figure 1?

The authors added the paragraph below to the Methods section, p. 13 (red).

“Background counts coming from other sources than fluorescence, such as dark counts and detector afterpulsing, appear as an offset in the histogram because of their uniform probability distribution at the time scale of the histogram (25 ns). As a result, bins in which the number of photons approaches this offset value can be considered to contain only background. Having access to the start-stop time with ps range resolution, the BrightEyes-TTM allows classifying every photon in either of two classes: (i) most likely background or (ii) possibly fluorescence. In the former case, the photon can be completely removed from the data; in the latter case, the filter function is used to add a weight to the photon depending on how likely it is that the photon comes from the sample.”

Please see major point #2 below for questions on using the number of SPAD array pixels as a proxy for focal volume.

The authors answer this question below.

Figure 5. This is another great experiment. I have a few questions about the results.

Note that in the new version of the manuscript Fig. 5 is now Fig. 6.

Please see major point #2 below for questions on using the number of SPAD array pixels as a proxy for focal volume.

The authors answer this question below.

It would be informative for the authors to provide the results for both the cytoplasm and nucleus.

The authors added the results for the nuclei (average diffusion coefficient of $(20 \pm 3) \mu\text{m}^2/\text{s}$) to the main text, p. 7, left column.

Where is the center of the beam focus along the axial extent of the cell?

Regarding the FLCS measurements, the beam focal plane was chosen visually to be axially in the center of the cell. In this context, it is essential to remember that the SPAD array effective size is 1.4 Airy Unit, thus it still acts as a pinhole, thus rejecting a substantial amount of out-of-focus background, which is important for fluorescence fluctuation experiments.

The cause of the variability in offsets and slopes for the dashed lines in (f) is not immediately clear. Can the authors please provide a clearer explanation of how the details of spot-variation analysis lead to the presented data?

The authors added a new figure, now Fig. 1, which describes schematically (panel c) how spot-variation FFS analysis is performed. Each measurement yields three time traces, corresponding to the central element trace, the sum3x3 trace, and the sum5x5 trace. From the corresponding correlation curves, the diffusion modality curve can be drawn. In Fig. 6 (e), the spatial variability in the curves for different positions in the cell is shown; in Fig. 6 (f), the temporal variability for a single position. In the former case, variations are caused by the different cellular environment of the nucleus with respect to the cytosol; in the latter case, by the heterogeneous environment of the cellular cytosol during physiological cellular processes. The authors explained this in more detail in the main text, p. 7.

Does the downward trend in $D_{\text{central}}/D_{5 \times 5}$ presented in panel (g) correspond with photobleaching? It does not appear so from the data in (c), I just wanted to confirm.

As the reviewer correctly pointed out, the photon count rate remained constant in all channels during the measurement, confirming that the change in the diffusion coefficient was not caused by photobleaching. The temporal variability in the measured diffusion coefficient can be attributed to the heterogeneous environment of the cellular cytosol during physiological cellular processes. A deeper understanding of the underlying processes is beyond the scope of the current work, but as already stated before, a work on which the FLFS approach is effectively used on a relevant biological application is in preparation.

2. Relationship of excitation beam (true ω_0) and detector derived ω_0 . In multiple areas of the manuscript, there are plots of data-derived quantities (concentration, diffusion constant, etc.) versus the excitation beam waist (ω_0) or the focal volume. The authors appear to calculate ω_0 based on the number of pixels on the SPAD array. How accurate is this? The intensity profile at the edges of the SPAD array may be significantly different from the center of the beam. Additionally, is the pixelation of the

SPAD array taken into account when calculating the effective ω_0 ? Uncertainties in ω_0 will alter the downstream analyses.

The reviewer points out an important aspect in FFS analyses; indeed, an accurate value of ω_0 is crucial to obtain correct quantitative results. This parameter is defined by a combination of the excitation and emission PSF of the system, of which the latter depends on which elements (i.e, central, sum3x3, sum5x5) are taken into account. There are three possible ways of getting information on ω_0 . (i) One can simulate the excitation and emission PSF taking into account various experimental parameters such as the excitation and emission wavelength and the NA of the objective. (ii) One can perform a calibration FFS experiment with a dye with a known diffusion coefficient and measure ω_0 . (iii) One can perform a calibration-free FFS experiment, such as circular scanning FCS, in which ω_0 can be extracted directly from the measurement itself. The authors chose the third approach since it is accurate (it takes into account all the experimental parameters that may influence ω_0 such as the pixelation of the detector and the laser power) and easy (since no extra calibration is needed).

The authors added panel (a) to Fig. 5 to show better how ω_0 was derived from the measurement and added the following reference to the paragraph on Fluorescence Lifetime Fluctuation Spectroscopy: Petrasek *et al.*, *Biophys. J.* **94**, 2001. This reference explains in more detail how scanning FCS can be used to perform quantitative measurements without a need for a standard.

The authors added the following phrase to the main text:

“In all cases, we fitted the data with a model assuming a 3D Gaussian focal volume with the diffusion time, the particle concentration, and ω_0 as fit parameters.”

More details on the reviewers’ concerns regarding the intensity profile near the detector edges can be found in the response below.

There is a second issue, as the beam's intensity profile is not truly a 3D Gaussian. Therefore, how much correction is required when accurately modeling the focal volume, e.g., using a Gauss-Lorentz function (for example, see doi: 10.1016/S0006-3495(01)76264-9)?

The authors agree with the reviewer that the assumption that the intensity profiles are a 3D Gaussian is not entirely accurate. Especially pixels near the edge of the detector will have a detection PSF that may significantly deviate from a 3D Gaussian. However, in the current work, the outer pixels are only used in combination with other pixels (sum 5x5). As shown before (Slenders *et al.*, *Light-Sci & Appl.*, 2021 and Slenders *et al.*, *Biophys. Rep.*, 2021), the deviation from a Gaussian is small enough in this case to justify the approximation without the need for additional corrections. For cross-correlation analyses of individual pixels, e.g., for cross-correlating the time trace of the central element with one of the corner elements, the intensity profile can indeed not be well approximated as a 3D Gaussian. This type of analysis was not done in the current work.

3. Overall vision/impact

In their conclusion, the authors state: "The principal aim of this work is to democratise single photon microscopy by giving any microscopy laboratory the possibility to update their existing LSM systems." It is unclear to me how the BrightEyes-TTM helps update existing LSM systems without adding a SPAD array. All of the use cases in the paper are focused on SPAD arrays detectors. At this point, it is unclear

how the community would adopt these methods without a SPAD array. I agree that the work presented here is an important step forward for the SPAD array community (and I wish we had a SPAD array to justify building a BrightEyes-TTM!). However, as presented, the lack of applicability to standard laser scanning microscopes does diminish the impact of the author's statement of purpose. Can the authors please elaborate on experiments or analyses that BrightEyes-TTM enables on a standard LSM that would not be possible otherwise?

SPAD array technology reached a high level of maturity and it will soon also cover a fundamental role in microscopy. In the context of laser scanning microscopy, SPAD array detectors have exciting perspectives, which the microscopy community immediately realized. However, as commented by the referee, these perspectives will become a reality only when SPAD array detectors will be available in any lab, in other words, when SPAD array detectors will become commercially available. As a group that dedicated substantial efforts to methods based on SPAD array detection, this need was clear from the beginning. Now, we are happy to see that two startups, namely Genoa Instruments from the IIT (Vicidomini G. holds shares of Genoa Instruments), and PI Imaging Technology from EPFL, started to sell SPAD array detectors for laser-scanning microscopy – PI Imaging is also working on a mega-pixel SPAD array detector for wide-field microscopy.

Thanks to this new scenario, we validated our BrightEyes-TTM with a commercial 7x7 SPAD array detector (PRISM Light detector LVDS outputs, Genoa Instruments). Specifically, we implemented FLISM imaging (Supplementary Fig. S10). Clearly, this experiment is fundamental toward the democratization of single-photon microscopy because it effectively proves the possibility of implementing this technique in any laboratory worldwide. However, we still want to maintain the focus of the paper on the open-source multi-channel time-tagging module.

We are aware that not all the labs would immediately have access to a SPAD array detector, but we believe that also these labs can contribute to this project and specifically validate the BrightEyes modules with more conventional detectors. We want to stress that this project is not only about the “hardware” part, but the project also includes a series of analysis software packages to analyze fluorescence lifetime and fluorescence fluctuation spectroscopy data. In short, our dream is to build an open-source community in the general topic of single-photon microscopy, which also includes fluorescence lifetime imaging/spectroscopy and fluorescence fluctuation spectroscopy.

We show that the BrightEyes-TTM, being an open-source multi-channel time-tagging module, allows any lab that has a standard LSM with single-element SPADs or PMTs and a pulsed-laser to perform lifetime measurements. Specifically, we performed several additional experiments.

- FLIM with the BrightEyes-TTM connected to a commercial microscope (Nikon A1 confocal microscope), Fig. S13;
- Fluorescence lifetime measurement with the BrightEyes-TTM connected to a custom-built setup with a commercial single photon detector (SPD-050-CTC-FC, Micro Photon Devices), Fig. S14.

Minor

1. When constructing the filtering functions for the correlation curves (page 12), the authors state: "Only photons in time bins between the peak of the histogram and about 10 ns later were included. In this way, photons that are likely to not originate from the sample were removed. "

Could the authors please explain the source and number of the excluded photons? Are these actual photon events, detector noise, or electronics noise?

The main sources of excluded events are dark counts and detector afterpulsing; both phenomena are not (directly) coming from actual photons. Dark counts can be caused by thermally triggered avalanches. Afterpulsing happens when a carrier gets trapped by deep energy levels of the semiconductor during a previous (photon-triggered) avalanche and is released some random time later. Both phenomena create an offset in the start-stop histogram, meaning that “photons” that were counted at start-stop times that are much larger than the fluorescence decay are most likely not coming from the sample but are dark counts or detector afterpulsing. The authors changed the word *photons* to *counts* in both sentences to clarify this statement.

The number of counts that were removed this way depends on the afterpulsing probability, the dark count rate of each pixel, and the fluorescence count rate. We removed a minimum 24% and a maximum 53% of the counts.

2. Shortly after the above statement, the authors further assert: "Then, for each photon, a weight was assigned equal to the value of the fluorescence filter function at the corresponding photon start-stop time. E.g., photons that were detected directly after the laser pulse were assigned a higher weight than photons detected some time later, since the probability of a photon being a fluorescence photon decreases with increasing start-stop time." Is this true in heterogeneous samples? The lifetime of both fluorescence dyes and proteins is highly sensitive to the local environment. I understand that having access to this data improves over prior capture cards. The increased data quality offers an opportunity to explain observations like this one.

Indeed, in biological samples, the lifetime cannot always be approximated as a single exponential decay and may vary locally. The authors applied the filter functions only to the FLFS data for the beads, showing that filtering is necessary to get the correct amplitude of the autocorrelation function while the diffusion coefficient (and lifetime) remains unaffected by the filter. In the beads sample, the start-stop time histograms could be well fitted with a single exponential decay. In more complex environments such as cells, this may not be the case. For this reason, the FLFS data in cells was not filtered, and only the diffusion times and lifetimes were analyzed, two parameters that do not require filtering. However, in single-point FLFS, the dynamics are probed locally, meaning that one does not have to account for the spatial heterogeneity, and one could simply use a different model (e.g., two-component exponential decay) to fit the data before filtering. Since the current work focusses on the data acquisition platform, the authors did not include this analysis in the manuscript. The authors updated the *Fluorescence correlation calculation and analysis* paragraph in the *Methods* section to make it clear that the filter functions were applied only to the beads sample:

“To obtain the filtered correlation curves for the beads sample, the same procedure was followed, except that a weight was assigned to each photon based on its start-stop time.”

3. Throughout the manuscript, there are many colloquial language usages, including more than one "In a nutshell,...". Please edit down and rewrite to communicate the essential points without such expressions.

The authors made the necessary changes.

--Douglas Shepherd

REVIEWERS' COMMENTS

Reviewer #1 (Remarks to the Author):

[NB: The previous comments and correction requests are not going to be repeated.]

I would like to thank the Authors for their extensive revision effort. A few points here and there still need to be clarified before final publication:

- A new, commercially available 7x7 SPAD array has been added to the analysis. While this is welcome, I have the feeling that the presence of 3 set-ups might lead to some confusion. I would therefore suggest to clearly specify which results have been obtained with which detector/measurement configuration.
- As an example, the "Results" section (p. 2 bottom right) starts by mentioning a 25 channel version for the current release, and 49 channels for next. Some readers might therefore wonder how the 7x7 SPAD array detector is read out. Similarly for the saturation photon rate mentioned at p. 3, bottom right, or the 5x5 SPAD array being mentioned at p. 4 (right below Fig. 2) but not the 7x7 one.

DETAILED COMMENTS:

- Results:

- o FLISM, p. 6: "Nonetheless, our BrightEyes-TTM can be used by upgrading the FPGA-firmware and by substituting the I/Os daughter card with a new one designed for the commercial 7x7 SPAD array detectors." -> it is not immediately clear if this has already been done or not, and if yes, on how many channels.

- Discussion:

- o P. 7: "All commercial TTMs require multiple synchronised modules to support 25 channels." -> this is very likely correct, but you might want to give it a final check.

- o P. 8: "Thus, this approach is not practically scalable to the large (megapixel) SPAD cameras requested by wide-field microscopy (34). Here, typically an application-specific integrated circuit (ASIC) TDC is implemented directly on each SPAD element (35), and..." -> very large SPAD cameras with timing capabilities do often employ gating, or when implementing TDCs, these are featured at the chip or cluster level (i.e. covering a certain number of pixels). True pixel-level TDCs are very resource-intensive.

- Bibliography:

- o Some caps are still missing, e.g. spad -> SPAD, Cmos -> CMOS, etc

- TYPOS (non-exhaustive list): please recheck carefully the English of the new additions.

- Supplementary:

- o Some modifications were not clearly tagged, and as such a bit hard to follow.

- o Bibliography (p. 3): Some caps issues in ref. [5], and perhaps others.

- o Fig. S8: I would suggest to specify with which detector this data was taken, even if it might represent a repetition. Same for all other data where there could be ambiguity, e.g. Fig. S9, etc.

- o Fig. S10: it is not immediately clear that this is the 7x7 SPAD array detector.

- o Fig. S14: is the \$PD-050-CTC-FC the same as mentioned in the main document?

- o Fig. S23: is the "SPAD array detector" shown in the bottom left part of the image representative of the 5x5 as well as the 7x7?

- o Fig. S29: the 2nd peak at around 100 ps might warrant some comments.

o Table S2: thank you for this extensive addition. I would suggest to also add the number of channels to make it complete.

- TYPOS (non-exhaustive list): Fig. S7: "immediately translate", "ofthe".
-

Reviewer #2 (Remarks to the Author):

The authors have satisfyingly addressed all my comments. Therefore, I'm happy to recommend publication of the article as is.

Reviewer #3 (Remarks to the Author):

The authors have addressed my concerns in their revised manuscript. Thank you for the thorough and thoughtful revision.

--Douglas Shepherd

The BrightEyes-TTM: an Open-Source Time-Tagging Module for Democratizing Single-Photon Microscopy

REVIEWERS' COMMENTS

Reviewer #1 (Remarks to the Author):

[NB: The previous comments and correction requests are not going to be repeated.] I would like to thank the Authors for their extensive revision effort. A few points here and there still need to be clarified before final publication:

The authors thank the reviewer for the positive feedback and the minor comments that still had to be corrected. The authors made the necessary changes, see the answers below for more details, and hope that the new version of the manuscript will be accepted for publication.

- **A new, commercially available 7x7 SPAD array has been added to the analysis. While this is welcome, I have the feeling that the presence of 3 set-ups might lead to some confusion. I would therefore suggest to clearly specify which results have been obtained with which detector/measurement configuration.**

The authors added in the caption of the figures with which setup and detector each measurement was performed.

- **As an example, the “Results” section (p. 2 bottom right) starts by mentioning a 25 channel version for the current release, and 49 channels for next. Some readers might therefore wonder how the 7x7 SPAD array detector is read out. Similarly for the saturation photon rate mentioned at p. 3, bottom right, or the 5x5 SPAD array being mentioned at p. 4 (right below Fig. 2) but not the 7x7 one.**

The authors now mention explicitly (p. 6 top left) that in this work only the 5x5 central area of the 7x7 detector is read out.

DETAILED COMMENTS:

* Results:

- o FLISM, p. 6: **“Nonetheless, our BrightEyes-TTM can be used by upgrading the FPGA-firmware and by substituting the I/Os daughter card with a new one designed for the commercial 7x7 SPAD array detectors.” -> it is not immediately clear if this has already been done or not, and if yes, on how many channels.**

See previous comment. The authors now mention explicitly (p. 6 top left) that in this work the 7x7 SPAD array detector has been used but that only the 5x5 central area of the 7x7 detector is read out.

- Discussion:
o P. 7: **“All commercial TTMs require multiple synchronised modules to support 25 channels.” -> this is very likely correct, but you might want to give it a final check.**

The authors checked this.

- o P. 8: **“Thus, this approach is not practically scalable to the large (megapixel) SPAD cameras requested by wide-field microscopy (34). Here, typically an application-specific integrated circuit (ASIC) TDC is**

implemented directly on each SPAD element (35), and..." -> very large SPAD cameras with timing capabilities do often employ gating, or when implementing TDCs, these are featured at the chip or cluster level (i.e. covering a certain number of pixels). True pixel-level TDCs are very resource-intensive.

The authors changed the paragraph to

Whilst the time-tagging module implements the most informative DAQ system strategy, it requires transferring (to the PC) and storing a consistent amount of data. Thus, this approach is not practically scalable to the large (megapixel) SPAD cameras requested by wide-field microscopy. Here, typically an application-specific integrated circuit (ASIC) TDC is implemented directly at the chip level for each SPAD element or cluster of SPAD elements, and the temporal information is pre-processed to reduce the amount of data to transfer.

• **Bibliography:**

o Some caps are still missing, e.g. spad -> SPAD, Cmos -> CMOS, etc

The authors checked the bibliography.

• **TYPOS (non-exhaustive list): please recheck carefully the English of the new additions.**

The authors checked the English throughout the whole paper as well as the Supplementary Information.

• **Supplementary:**

o Some modifications were not clearly tagged, and as such a bit hard to follow.

The authors uploaded two versions of the manuscript, one of which has the changes colored in red.

o **Bibliography (p. 3): Some caps issues in ref. [5], and perhaps others.**

The authors carefully checked the bibliography.

o **Fig. S8: I would suggest to specify with which detector this data was taken, even if it might represent a repetition. Same for all other data where there could be ambiguity, e.g. Fig. S9, etc.**

All relevant figures now mention which detector/setup was used.

o **Fig. S10: it is not immediately clear that this is the 7x7 SPAD array detector.**

All relevant figures now mention which detector/setup was used.

o **Fig. S14: is the \$PD-050-CTC-FC the same as mentioned in the main document?**

The authors thank the reviewer for noticing this mistake. The detector in the main document was changed to \$PD-050-CTC-FC.

o **Fig. S23: is the "SPAD array detector" shown in the bottom left part of the image representative of the 5x5 as well as the 7x7?**

No, only the 5x5. The caption was changed to clear this up.

o **Fig. S29: the 2nd peak at around 100 ps might warrant some comments.**

This peak is due to intrinsic FPGA irregularities. We refer now to this figure in the paragraph *bin-by-bin calibration* in Suppl. Note 1.

o Table S2: thank you for this extensive addition. I would suggest to also add the number of channels to make it complete.

The authors added an extra column to the table indicating the number of channels.

• TYPOS (non-exhaustive list): Fig. S7: “immediately translate”, “ofthe”.

The authors carefully read the Supplementary Information and corrected several typos.

Reviewer #2 (Remarks to the Author):

The authors have satisfyingly addressed all my comments. Therefore, I'm happy to recommend publication of the article as is.

The authors thank the reviewer for the positive feedback.

Reviewer #3 (Remarks to the Author):

The authors have addressed my concerns in their revised manuscript. Thank you for the thorough and thoughtful revision.

--Douglas Shepherd

The authors thank the reviewer for the positive feedback.